# Early prediction of clinical response to checkpoint inhibitor therapy in human solid tumors through mathematical modeling

Joseph D Butner[1†], Geoffrey V Martin[2†], Zhihui Wang[1,3*†], Bruna Corradetti[4,5], Mauro Ferrari[4], Nestor Esnaola[6], Caroline Chung[2], David S Hong[7], James W Welsh[2], Naomi Hasegawa[8], Elizabeth A Mittendorf[9], Steven A Curley[10], Shu-Hsia Chen[11], Ping-Ying Pan[11,12], Steven K Libutti[13,14], Shridar Ganesan[13], Richard L Sidman[15], Renata Pasqualini[16,17], Wadih Arap[16,18], Eugene J Koay[2*†], Vittorio Cristini[1,3]

[1]Mathematics in Medicine Program, Houston Methodist Research Institute, Houston, United States; [2]Department of Radiation Oncology, The University of Texas M.D. Anderson Cancer Center, Houston, United States; [3]Department of Imaging Physics, The University of Texas M.D. Anderson Cancer Center, Houston, United States; [4]Department of Nanomedicine, Houston Methodist Research Institute, Houston, United States; [5]Swansea University Medical School, Singleton Park, Swansea, United Kingdom; [6]Department of Surgery, Houston Methodist Cancer Center, Houston, United States; [7]Department of Investigational Cancer Therapeutics, The University of Texas M.D. Anderson Cancer Center, Houston, United States; [8]University of Texas Health Science Center (UTHealth), McGovern Medical School, Houston, United States; [9]Breast Oncology Program, Dana Farber/Brigham and Women's Cancer Center, Boston, United States; [10]Michael E. DeBakey Department of Surgery, Baylor College of Medicine, Houston, United States; [11]Immunotherapy Research Center, Houston Methodist Research Institute, Houston, United States; [12]Cancer Center, Houston Methodist Research Institute, Houston, United States; [13]Rutgers Cancer Institute of New Jersey, New Brunswick, United States; [14]Department of Surgery, Rutgers Robert Wood Johnson Medical School, New Brunswick, United States; [15]Department of Neurology, Harvard Medical School, Boston, United States; [16]Rutgers Cancer Institute of New Jersey, Newark, United States; [17]Division of Cancer Biology, Department of Radiation Oncology, Rutgers New Jersey Medical School, Newark, United States; [18]Division of Hematology/Oncology, Department of Medicine, Rutgers New Jersey Medical School, Newark, United States

*For correspondence:
zwang@houstonmethodist.org
(ZW);
EKoay@mdanderson.org (EJK)

†These authors contributed
equally to this work

Competing interest: See page
17

Reviewing Editor: Caigang Liu,
Shengjing Hospital of China
Medical University, China

## Abstract

**Background:** Checkpoint inhibitor therapy of cancer has led to markedly improved survival of a subset of patients in multiple solid malignant tumor types, yet the factors driving these clinical responses or lack thereof are not known. We have developed a mechanistic mathematical model for better understanding these factors and their relations in order to predict treatment outcome and optimize personal treatment strategies.

**Methods:** Here, we present a translational mathematical model dependent on three key parameters for describing efficacy of checkpoint inhibitors in human cancer: tumor growth rate ($\alpha$),

DOI: https://doi.org/10.7554/eLife.70130

tumor-immune infiltration ($\Lambda$), and immunotherapy-mediated amplification of anti-tumor response ($\mu$). The model was calibrated by fitting it to a compiled clinical tumor response dataset (n = 189 patients) obtained from published anti-PD-1 and anti-PD-L1 clinical trials, and then validated on an additional validation cohort (n = 64 patients) obtained from our in-house clinical trials.

**Results:** The derived parameters $\Lambda$ and $\mu$ were both significantly different between responding versus nonresponding patients. Of note, our model appropriately classified response in 81.4% of patients by using only tumor volume measurements and within 2 months of treatment initiation in a retrospective analysis. The model reliably predicted clinical response to the PD-1/PD-L1 class of checkpoint inhibitors across multiple solid malignant tumor types. Comparison of model parameters to immunohistochemical measurement of PD-L1 and CD8+ T cells confirmed robust relationships between model parameters and their underlying biology.

**Conclusions:** These results have demonstrated reliable methods to inform model parameters directly from biopsy samples, which are conveniently obtainable as early as the start of treatment. Together, these suggest that the model parameters may serve as early and robust biomarkers of the efficacy of checkpoint inhibitor therapy on an individualized per-patient basis.

**Funding:** We gratefully acknowledge support from the Andrew Sabin Family Fellowship, Center for Radiation Oncology Research, Sheikh Ahmed Center for Pancreatic Cancer Research, GE Health-care, Philips Healthcare, and institutional funds from the University of Texas M.D. Anderson Cancer Center. We have also received Cancer Center Support Grants from the National Cancer Institute (P30CA016672 to the University of Texas M.D. Anderson Cancer Center and P30CA072720 the Rutgers Cancer Institute of New Jersey). This research has also been supported in part by grants from the National Science Foundation Grant DMS-1930583 (ZW, VC), the National Institutes of Health (NIH) 1R01CA253865 (ZW, VC), 1U01CA196403 (ZW, VC), 1U01CA213759 (ZW, VC), 1R01CA226537 (ZW, RP, WA, VC), 1R01CA222007 (ZW, VC), U54CA210181 (ZW, VC), and the University of Texas System STARS Award (VC). BC acknowledges support through the SER Cymru II Programme, funded by the European Commission through the Horizon 2020 Marie Skłodowska-Curie Actions (MSCA) COFUND scheme and the Welsh European Funding Office (WEFO) under the European Regional Development Fund (ERDF). EK has also received support from the Project Purple, NIH (U54CA210181, U01CA200468, and U01CA196403), and the Pancreatic Cancer Action Network (16-65-SING). MF was supported through NIH/NCI center grant U54CA210181, R01CA222959, DoD Breast Cancer Research Breakthrough Level IV Award W81XWH-17-1-0389, and the Ernest Cockrell Jr. Presidential Distinguished Chair at Houston Methodist Research Institute. RP and WA received serial research awards from AngelWorks, the Gillson-Longenbaugh Foundation, and the Marcus Foundation. This work was also supported in part by grants from the National Cancer Institute to SHC (R01CA109322, R01CA127483, R01CA208703, and U54CA210181 CITO pilot grant) and to PYP (R01CA140243, R01CA188610, and U54CA210181 CITO pilot grant). The funders had no role in study design, data collection and analysis, decision to publish, or preparation of the manuscript.

## Editor's evaluation

A mathematical model was established for predicting immunotherapy efficacy in this work. With three convenient available clinical parameters, the model has exhibited considerable predictive capacity with stable performance across several tumor types. It may show great promise in selecting participants for prospective trials and guiding targeted application of immunotherapy in cancer patients.

## Introduction

Recent advances in the understanding of immunological pathways responsible for antibody- and/or cell-mediated destruction of tumors have led to the development of unique cancer therapeutics in recent years, leading to markedly improved survival in the setting of previously intractable metastatic melanoma, bladder, kidney, lung, and head and neck cancers, among several other human solid tumor types (*Borghaei et al., 2015*; *Robert et al., 2015a*; *Robert et al., 2015b*). In particular, one of the most successful clinical applications of immune checkpoint inhibitors includes antibodies directed

against the programmed death protein 1 (PD-1) pathway, which inhibits cellular immune killing of cancer cells via complementary binding of tumor expressed programmed death ligand 1 (PD-L1) to PD-1 on immune cells (*Pardoll, 2012*). As a still emerging yet quite compelling immunotherapy approach in contemporary cancer medicine, targeting of immune checkpoints is being extensively investigated in ongoing and upcoming clinical trials aimed at unleashing T cell activity, augmenting immune recognition against tumor metastases, and boosting immune memory for long-lasting clinical remission post-treatment (*Postow et al., 2015*; *Le et al., 2017*). While the remarkable potential for checkpoint inhibitors in treating cancer is unequivocally exciting, the combined clinical trial experience has shown that durable effective treatment outcomes occur only in a limited subset of patients (*Sharma et al., 2017*). Alas, some cancer types are presumed to be minimally immunogenic, resulting in little or no response to this treatment strategy (*Brahmer et al., 2012*). An accumulating body of evidence has established that certain immunological features, including T cell exhaustion (e.g., Tim-3) and exclusion (*Jerby-Arnon et al., 2018*), senescence markers (*Moreira et al., 2019*) such as CD57, or immune incompetence, exhaustion, or premature senescence (e.g., loss of CD27) (*Riaz et al., 2017*; *Vallejo, 2005*), could perhaps reflect or even predict sensitivity and resistance to checkpoint inhibitor-based cancer immunotherapy. However, early attempts at identification of specific pathological biomarkers to predict immunotherapy response, including transcriptomic rubrics (*Auslander et al., 2018*), machine learning algorithms (*Johannet et al., 2021*), genomic approaches (*Cormedi et al., 2021*) such as tumor mutational burden (*Duffy and Crown, 2019*), among others (*Pilard et al., 2021*), have thus far shown somewhat inconsistent results, challenged further by the inherent molecular, cellular, and biophysical diversity of human tumors (*Teng et al., 2015*; *Tumeh et al., 2014*; *Carbognin et al., 2015*). Instead, most currently applied parameters merely document tumor responses that have already occurred as opposed to predicting it a priori; these include the standard Response Evaluation Criteria in Solid Tumor (RECIST) v1.1 (*Eisenhauer et al., 2009*) and even the newer proposed response assessment rubrics specific for immunotherapy (iRECIST). Considering the sheer complexity of the biological interactions between the immune system and the tumor microenvironment, one could reason that the introduction of more sophisticated mathematical analytic techniques would hopefully have potential to enhance the qualitative and quantitative understanding of such interactions and to improve malignant tumor treatment with checkpoint inhibitors, ultimately supporting the development of a predictive clinical tool that could either minimize or overcome this clear unmet need of cancer medicine.

In previous work, mechanistic mathematical models of immunomodulatory interventions for cancer control utilizing coupled ordinary differential equations (ODEs) have already allowed the prediction of tumor response after Bacillus Calmette–Guérin (BCG) administration in superficial transitional cell carcinoma of the urinary bladder and the serum prostate-specific antigen response after prostate cancer vaccine administration with considerable accuracy (*Bunimovich-Mendrazitsky et al., 2016*; *Bunimovich-Mendrazitsky et al., 2007*; *Kronik et al., 2010*). Moreover, other investigators have shown that mechanistic models of interleukin-21 (IL-21) therapy schedules based on tumor mass and antigenic properties can predict growth patterns of multiple tumor types in patients receiving personalized doses of IL-21 (*de Pillis et al., 2005*). In another notable work, mathematical modeling of neoantigen fitness based on cancer population evolution and antigen recognition potential was able to predict tumor response and patient survival following treatment with checkpoint blockade therapy, as validated in both melanoma patients treated with anti-CTLA4 and in lung cancer patients treated with anti-PD-1 therapies (*Łuksza et al., 2017*). Finally, modeling of chemotherapy and targeted drug therapy by our group with similar mathematical techniques was able to accurately reproduce entire dose–response curves of tumor cell kill with untargeted cytotoxic and ligand-directed agents (*Pascal et al., 2013a*; *Hosoya et al., 2016*; *Dogra et al., 2018*; *Goel et al., 2019*; *Dogra et al., 2020b*). These examples highlight the previous success of mathematical models to qualitatively or quantitatively discern the underlying biologically and physically relevant, often nonlinear processes present in cancer, which may otherwise be missed, and help optimize treatment delivery approaches.

The innovation and scope of the present study is to demonstrate how standard clinical measures, including radiological imaging assessments and immunohistochemical (IHC) analysis of tissue samples, may inform a mechanistic mathematical model to elucidate patient and tumor-specific attributes that would likely benefit from the application of checkpoint inhibitor-based therapy. Our model parameters, which can be determined in multiple ways (e.g., from early time point imaging and/or

histopathology data), present a clear advantage over the standard measures currently used in the clinic for predicting treatment outcome and patient survival. The mathematical model presented in this report is based on an extensive series of prior methodological reports (*Pascal et al., 2013a*; *Hosoya et al., 2016*; *Dogra et al., 2018*; *Goel et al., 2019*; *Dogra et al., 2020b*; *Das et al., 2013*; *Pascal et al., 2013b*; *Wang et al., 2016*; *Koay et al., 2014*; *Frieboes et al., 2015*; *Wang et al., 2015*; *Brocato et al., 2018*; *Cristini et al., 2017*; *Dogra et al., 2019*; *Brocato et al., 2019*; *Dogra et al., 2020a*; *Goel et al., 2020*; *Anaya et al., 2021*), but with rigorous emphasis on parameters related to the cancer-immune response to the treatment with checkpoint inhibitors in the setting of patients with solid tumors. In particular, we have developed a model that contains key mechanistic biological and physical processes involved in checkpoint inhibitor therapy, which has been objectively validated against imaging data of patient tumor burden and measured response, defined by both change in total tumor burden and patient survival (*Butner et al., 2020*). To date, this modeling work has only examined tumor response on a bulk scale, without confirming the mechanistic links between key model parameters and the underlying biology they describe. In this work, we demonstrate how model parameters may be informed by using cell-scale IHC analysis of biopsied tissues, which are available as early as at the start of treatment, for reliable prediction of therapeutic response. Thus, this represents a key step towards clinical translation (i) by validating prior results in additional patient cohorts and (ii) demonstrating how IHC may be used with the model to predict patient response at times closer to the start of treatment. The predictions obtained through this model were compared against available retrospective clinical data from published trials with monoclonal antibodies blocking the PD-1/PD-L1 pathway for validation, thereby focusing on a quantitative relationship between tumor response and its underlying mechanisms. Taken together, these retrospective results indicate that our predictive mathematical model with the class of PD-1/PD-L1 checkpoint inhibitors has translational merit and that it should therefore be evaluated as an integral predictive biomarker in the setting of carefully designed prospective cancer investigational trials.

## Methods
### Mathematical model
The final mathematical model we used for comparison with clinical data is a single ODE, which determines changes in relative tumor mass over time after initiation of anti-PD-1 therapy. It is a simplified

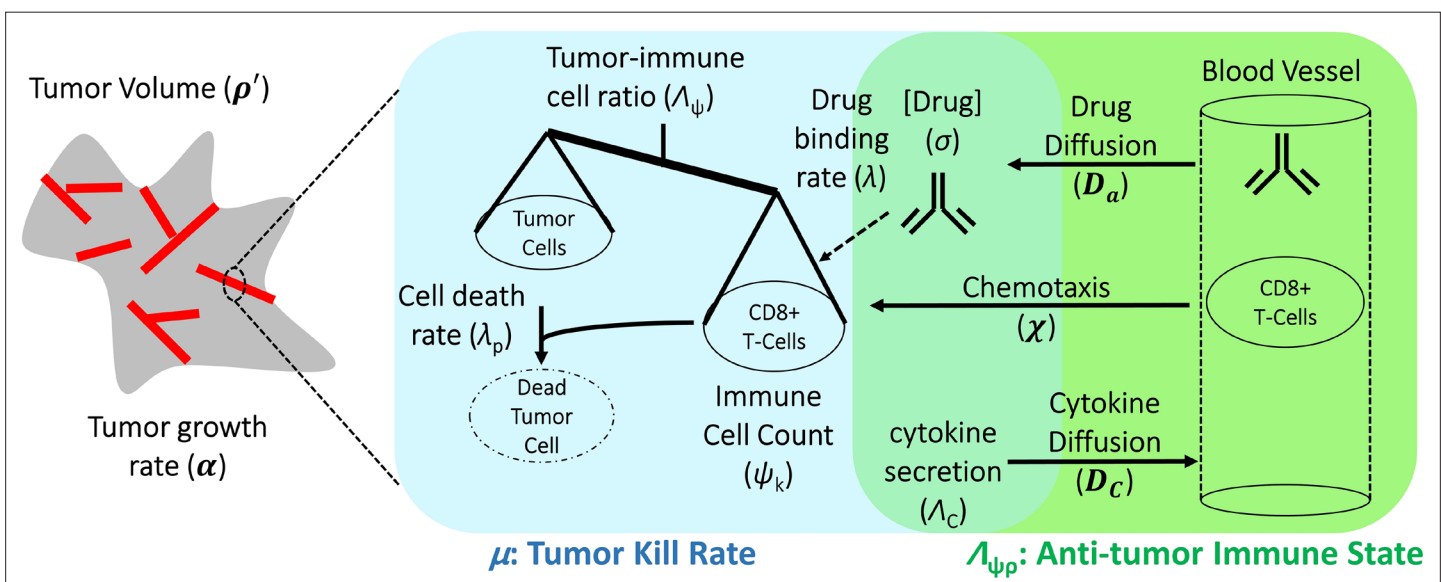

**Figure 1.** Schematic representation of biological mechanisms included in the mathematical model. These processes are described by four partial differential equations, which are solved to obtain Equation (1). Briefly, the checkpoint inhibitor enters the tumor via diffusion ($D_a$) leading to time-dependent drug concentration (σ), which then binds to the conjugate receptor on immune cells at rate $\lambda$. Immune cells ($\psi_k$) are drawn into the tumor microenvironment via cytokine-mediated chemotaxis ($\chi$), resulting in immune checkpoint inhibitor-mediated cancer cell kill at rate $\lambda_p$. The full mathematical model derivation and its underlying assumptions are provided in a recent modeling and analysis report (*Butner et al., 2020*).

and user-friendly version originated from a complex set of partial differential equations (PDEs), which takes into account spatial relationships within the tumor microenvironment. By reducing the model to this closed-form solution, we present the model in a form that combines related biological processes (see *Figure 1*) into only a few easy-to-interpret values, while also ensuring the ability to obtain unique solutions when only minimal data are available. The full mathematical derivation has been demonstrated elsewhere (*Butner et al., 2020*) and also in Appendix 1, but its underlying cancer biology is also schematically depicted in *Figure 1*. It is ultimately represented by

$$\rho' = \frac{\rho_\infty}{1 - (1 - \rho_\infty)e^{-(\alpha - \mu + \mu \cdot \Lambda) \cdot t}} \tag{1}$$

where $\rho'$ is the normalized tumor mass, $t$ is time, $\alpha$ is a proliferation constant of tumor cells, $\Lambda$ is the ratio of cancer cells ($\sigma$) to effector immune cells multiplied by the ability of immune cells ($\psi_k$) to kill cancer cells (namely, the anti-tumor immune state), $\mu$ is the effect of immunotherapy on the ability of immune cells to kill cancer cells, and $\rho_\infty$ represents the final long-term tumor burden, which may be calculated as

$$\rho_\infty = 1 + \frac{\alpha - \mu}{\mu \cdot \Lambda} \tag{2}$$

In words, *Equation (1)* describes the time-course tumor burden under immunotherapy intervention as a function of three key parameters $\alpha$, $\Lambda$, and $\mu$, which represent the ability of malignant tumor cells to grow, the ability of immune cells to kill cancer cells within the tumor environment, and the potential efficacy of checkpoint inhibitor-based immunotherapy treatment. *Equation (2)* states that long-term tumor burden is a function of the relationship between tumor growth and kill rates [the greater of which sets the sign of response, i.e., progressive disease ($\alpha > \mu$) or favorable clinical response ($\alpha < \mu$)], as scaled by the parameter $\mu \cdot \Lambda$. These three parameters are intrinsically abstract terms, which have multiple components in themselves. $\Lambda$ is dependent on the density of malignant tumor versus immune cells at the initiation of immunotherapy ($t = 0$) and on the ability of each immune cell to kill tumor cells. *Equation (1)* assumes the following five assumptions: (i) tumor mass grows exponentially with rate $\alpha$ in the absence of immune cell killing; (ii) cancer cells are killed by one type of net effector immune cell (termed $\psi_k$); (iii) the influx of immune cells and the concentration of checkpoint inhibitor within the tumor environment reaches a drug steady state on time scales far faster (i.e., from hours to days) than clinical response (i.e., from weeks to months); (iv) all cancer cells are within an average intratumoral movement path length of effector immune cells, and the effect of cytokines in stimulating immune cell function/movement is small, thus spatial diversity effects are negligible in this particular case; and (v) the site of action of the checkpoint inhibitor takes place at the interface between the tumor cell and immune cell, with blocking of one of the immune checkpoint receptors, either on the cancer or on the immune cell, which are deemed necessary and sufficient for blockade of the immune inhibitory pathway.

**Table 1.** Clinical trials with checkpoint inhibitors used to fit the mathematical model and derived values of tumor proliferation constant, $\alpha$.

| Reference | Tumor type histopathology | Checkpoint inhibitor monoclonal antibody | Constant $\alpha$ (days$^{-1}$) | Calculated tumor doubling time ($\alpha^{-1}$, days) |
|---|---|---|---|---|
| *Le et al., 2015* | CRC | Pembrolizumab (anti-PD1) | 0.0622 | 11 |
| *Powles et al., 2014* | UCC | Atezolizumab (anti-PD-L1) | 0.016 | 43 |
| *Antonia et al., 2015* | SCLC | Nivolumab (anti-PD1) | 0.014 | 50 |
| *Topalian et al., 2012* | MM | Nivolumab (anti-PD1) | 0.0069 | 100 |
| *Borghaei et al., 2015* | NSCLC | Nivolumab (anti-PD1) | 0.0069 | 100 |
| *Motzer et al., 2015* | RCC | Nivolumab (anti-PD1) | 0.0034 | 204 |

CRC = colorectal carcinoma. UCC = urothelial cell carcinoma. SCLC = small cell lung cancer. MM = malignant melanoma. NSCLC = non-small lung cancer. RCC = renal cell carcinoma.

While one must recognize that although these assumptions certainly represent a gross biological generalization of the true tumor-immune microenvironment complexity, they enable direct comparison with clinical response data, in which measurements of all the parameters required for a full description of immune system reaction to tumors are often not readily available in conventional clinical settings. In this retrospective mechanistic analysis, we start with a simple biophysical model to validate clinical findings and the mechanistic links between high-level model parameters and the underlying biology they describe, with the possible addition of further refining variables as an aspirational goal for future prospective translational or clinical cancer studies. The mathematical simulations and procedures conducted are fully described in Appendix 1 (*Appendix 1—figures 1 and 2*); finally, we also address the implications of these arbitrary choices in 'Discussion'.

## Clinical evaluation and fitting of the mathematical model

To evaluate *Equation (1)* with clinical data, we initially focused on clinical trials with the class of checkpoint inhibitors that specifically target the PD-1/PD-L1 pathway. In this analysis, tumor volumes over time measurements were obtained from six published clinical trial reports (*Borghaei et al., 2015*; *Antonia et al., 2015*; *Le et al., 2015*; *Motzer et al., 2015*; *Powles et al., 2014*; *Topalian et al., 2012*). As an initial retrospective proof of concept, five clinical trials of anti-PD-1 therapy plus another clinical trial of anti-PD-L1 treatment (*Table 1*) were included, with the combined dataset representing several common solid tumor pathology types (n = 189 cancer patients); this dataset represents our model calibration cohort. An online tool (Web Plot Digitizer [*Rohatgi, 2010*]) served to extract the individual patient tumor response data over time from the published clinical trials, and subsequently *Equation (1)* was fit to these individual and collective datasets to derive the parameters $\alpha$, $\Lambda$, and $\mu$. A

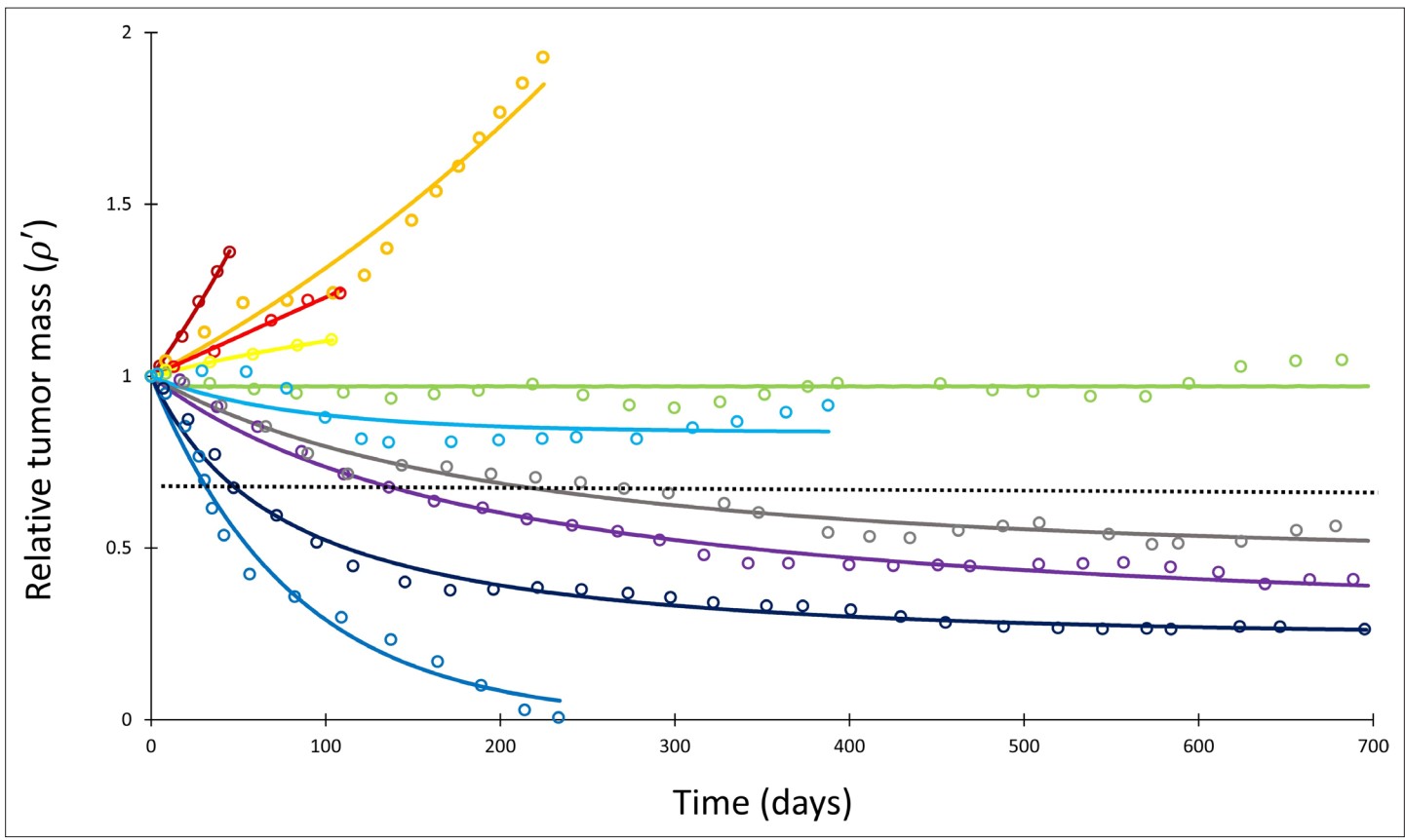

**Figure 2.** Mathematical model fit to individual responses to immune checkpoint inhibition. Open circles represent data points of clinical response in 10 patients extracted from *Topalian et al., 2012*, while solid lines represent best curve fits of *Equation (1)* to those data (with $\alpha^{-1}$ = 144 days). Each color represents a different patient. Immunotherapy was begun at $t = 0$, and tumor volume was designated as the relative change in volume from $t = 0$ (i.e., tumor volume of 1 at $t = 0$). The dashed line depicts the cutoff used for classifying patients deemed as responders (partial or complete response) versus nonresponders (stable disease or disease progression) according to the RECIST v1.1 criteria.

unique fit (and thus set of parameter values) was obtained for the data from each individual patient, and more details on this procedure may also be found in *Butner et al., 2020*.

The fitting procedure was performed in two sequential steps. First, the fastest progressing patient in each trial was fit to a simple exponential function to estimate $\alpha$ for each cancer histopathology (i.e., approximation for uninhibited tumor doubling time). Although this approximation represents a limiting factor in the study, it was necessary due to lack of pretreatment tumor measurement in the calibration cohort, which would allow for more accurate estimation of the growth kinetic without clinical intervention. As presented, $\alpha$ may represent the fastest tumor growth potential in the cohort and thus and overestimate the true average growth kinetic; however, because it likely represents the patient with the least response to treatment in each cohort, we take it to be a reasonable approximation of tumor growth kinetic unaffected by treatment. Second, this cancer-specific $\alpha$ was used to determine the remaining parameters, $\Lambda$ and $\mu$, for each individual patient by fitting *Equation (1)* to time-course tumor burden data (*Figure 2* and *Appendix 1—figure 1*). The curve fitting was performed by using nonlinear least squares and the Mathematica function NonLinearModelFit (*Wolfram Research I, 2017*). The workflow for deriving the model parameters from the clinical data is depicted in *Appendix 1—figure 1*.

The results from this model calibration were checked against an additional validation patient cohort from the University of Texas M.D. Anderson Cancer Center (MDACC). Briefly, data from patients (n = 64) with non-small cell lung cancer (NSCLC) treated with pembrolizumab (MK-3475) were obtained from the investigational study NCT02444741 (a total of 95 patients were obtained; however, 18 left the study after admission, 11 were removed due to lack of pretreatment measurements, 1 was excluded because all indexed lesions were treated with XRT, and 1 discontinued treatment but continued follow-up). Serial lung tumor measurements were taken on post-contrast chest CT images acquired with 2.5 mm slice thickness. All lesions that were previously indexed by the radiologist were included in order to best reproduce actual clinical practice (patients had a range of 1–12 indexed lesions; median, 3 indexed lesions). Lesion volumes were estimated as 3D spheres calculated from the geometric mean of long and short axes of each lesion at each follow-up, and then volumes were summed to generate a total volume for all indexed lesions at each follow-up time point. Because imaging data before the start of pembrolizumab treatment were available for each patient within this cohort, we were able to obtain a patient-specific pretreatment growth rate $\alpha$ for each patient between the imaging immediately preceding start of treatment and at the time of first immunotherapy treatment by using the exponential growth rate estimation $\rho(t) \approx e^{\alpha \cdot t}$ between tumor burdens measured via imaging collected before treatment and at start of treatment. These per-patient $\alpha$ values were then used in *Equation (1)* to obtain values for $\Lambda$ and $\mu$, as described above.

## Sensitivity analysis and model simulations

A sensitivity analysis was next performed on *Equation (1)* in two independent ways (*Appendix 1—figure 2*). First, each of the parameters $\Lambda$ and $\mu$ were individually altered by ±10% while holding the other parameters constant to simulate the change in relative tumor mass ($\rho'$). Second, individual values of $\rho'$ in the calibration cohort were varied by a uniform random variable by ±0.1 (i.e., 10%) and *Equation (1)* was re-fitted to these modified data to determine any changes in the derived values for $\Lambda$ and $\mu$. To determine stability of derived $\Lambda$ and $\mu$ values over time, we also truncated the $\rho'$ data at differing time points (namely, t < 30, 60, 120, 200 days), and subsequently re-fitted *Equation (1)* to these truncated data to determine $\Lambda$ and $\mu$. The correlation between the derived values of $\Lambda$ and $\mu$ from the truncated datasets and the full composite dataset was then determined by using Spearman correlation analysis. A simulated long-term RECIST v1.1 score was then determined for each patient by projecting the expected tumor burden at t = 700 days (chosen as a time point beyond the last reported follow-up for most of the patients) by using these values of $\Lambda$ and $\mu$ derived from the truncated fits in order to compare the predicted RECIST scores to the known retrospective scores. Finally, full model simulations were performed at different levels of $\alpha$, $\Lambda$, and $\mu$ to determine values of $\rho'$ based on input parameter variation.

## Validation of $\Lambda$ and $\mu$ by tumor-infiltrating immune cells and immunostaining

The parameters $\Lambda$ and $\mu$ derived from fitting the clinical response data were compared to pathological biomarkers from additional anti-PD-1 or PD-L1 clinical trials (**Borghaei et al., 2015**; **Robert et al., 2015b**; **Brahmer et al., 2012**; **Tumeh et al., 2014**; **Motzer et al., 2015**; **Powles et al., 2014**; **Topalian et al., 2012**; **Garon et al., 2015**; **Herbst et al., 2014**; **Kefford et al., 2014**; **Spira et al., 2015**; **Taube et al., 2014**; **Weber et al., 2015**), as shown in **Appendix 1—table 1**. Specifically, we investigated the ability of $\Lambda$ and $\mu$ to correlate with CD8+ tumor-infiltrating lymphocytes (TILs) and tumor PD-L1 expression, respectively. To compare the values of $\Lambda$ and $\mu$ derived from fitting the clinical response data over time with the trials assessing TILs and PD-L1 expression, the extracted cancer patients were categorized into objective responders versus nonresponders by applying RECIST v1.1 (**Eisenhauer et al., 2009**) note that resist categories determined by lesion diameter ($D$) or our volume estimation are mathematically comparable by $Volume = \frac{4}{3}\pi\left(\frac{D}{2}\right)^3$. Specifically, patients from all RECIST categories (**Eisenhauer et al., 2009**) were condensed into two groups for analysis, with patients who had ≥30% reduction in tumor burden (partial or complete response) grouped into the 'favorable' response group, and patients with either <30% disease reduction (stable disease or disease progression) into the 'unfavorable' response group. The last recorded time point from the response curve from each individual patient served to designate the RECIST v1.1 category as this provides the most accurate long-term treatment outcome.

After stratifying each patient into a RECIST v1.1 category, $\Lambda$ and $\mu$ were compared between responding and nonresponding patients by using a Wilcoxon rank sum test and compared to the literature either as a continuous variable ($\Lambda$) or by using thresholds ($\mu$). For comparison with biomarkers reported in the literature, $\Lambda$ was converted to an estimated intratumoral CD8+ T cell count (for details, the interested reader is referred to **Butner et al., 2020**) by assuming each CD8+ T cell would kill one tumor cell on average (the assumed mean 'fitness' of the immune cell population [**Mempel and Bauer, 2008**]), and that there were 5558 cells/mm$^2$ in the tumor microenvironment, as has been quantitatively measured in melanoma (**Erdag et al., 2012**). For comparison with PD-L1 staining, $\mu$ was converted from its raw numerical value to a percentage; no other scaling of the variable was performed as the number of cancer cells bound by anti-PD-1/PD-L1 therapy action is the dominant term in the integral specified in Equation S4 in **Butner et al., 2020**; this approximation and its implications are further examined in 'Discussion.' Assessment of objective response rates (ORRs) at standard thresholds of 1% and 5% PD-L1 staining (corresponding to $\mu$) were used in this study. Other alternative PD-L1 cutoffs present in some studies (**Table 1** and **Appendix 1—table 1**) were not included in this analysis due to the relatively small numbers of patients. The biomarker trials used for validation of these parameters are also presented in **Appendix 1—table 1**.

## Results

### Quantification of model parameters $\alpha$, $\Lambda$, and $\mu$

Measured tumor burden data over time from the total collective calibration patient pool (n = 189) after initiating checkpoint inhibitor-based therapy were extracted from the six reported clinical trials (**Table 1**). Of this population, 55 patients (29%) had objective responses by RECIST v1.1 criteria while 134 patients (71%) demonstrated stable/progressive disease. The derived tumor proliferation constant, $\alpha$, was determined to range from 0.0034 (tumor doubling time of ~200 days) in renal cell carcinoma to 0.0622 (tumor doubling time of ~11 days) in non-mismatch repair colon cancer with an average $\alpha$ of 0.018 and average tumor doubling time of ~85 days (**Table 1**). In the validation NSCLC cohort, 25 patients (39%) had objective response, and 39 patients (61%) demonstrated stable/progressive disease, with individual patient proliferation constants ($\alpha$) ranging from –0.0129 to 0.0602 (note that $\alpha$ < 0 indicates a subset of patients that had a shrinking tumor burden before start of therapy).

Having determined $\alpha$ for each cancer histopathology, the clinical response curves to checkpoint inhibitors for each tumor type were fit to **Equation (1)** to determine $\Lambda$ and $\mu$. The average root-mean-square error of fitting **Equation (1)** to the clinical data was only 0.4%. A sample exponential fit to extracted melanoma time-course data is shown (**Appendix 1—figure 1**, second panel, red curve). A sample of these curve fits for patients with melanoma from the trial by **Topalian et al., 2012** is

depicted in *Figure 2*. The derived fits from all 189 patient response curves in the calibration cohort yielded a mean ± standard error of the mean (SEM) value of $\Lambda$ of 0.714 ± 0.257 in patients with partial/complete response and 0.0995 ± 0.0264 in patients with stable/progressive disease (p=0.119; Wilcoxon two-tail), while the mean ± SEM value of $\mu$ was 0.054 ± 0.014 versus 0.013 ± 0.0012 in partial/complete response versus stable/progressive disease, respectively (p<0.001). In the validation NSCLC cohort, the mean ± SEM value of $\Lambda$ was 0.876 ± 0.102 in patients with partial/complete response and 0.0297 ± 0.469 in patients with stable/progressive disease (p<0.001), while the mean ± SEM value of $\mu$ was 0.0529 ± 0.00982 to –0.0064 ± 0.0032 in partial/complete response versus stable/progressive disease (p<0.001). These values are depicted in *Figure 3*. These results suggest that, while $\Lambda$ and $\mu$ give significantly separated classification ranges for partial/complete response versus stable/progressive disease, the specific value of the binary classification threshold is likely a function of the unique disease-drug combination used, as observed in our prior studies (*Butner et al., 2021*). This point was explored further through a 'leave-one-cancer-type-out' validation study within the calibration cohort, wherein one cancer type was removed from the calibration cohort and used as validation against the parameter ranges in the reduced calibration set obtained from *Borghaei et al., 2015*; *Antonia et al., 2015*; *Le et al., 2015*; *Motzer et al., 2015*; *Powles et al., 2014*; and *Topalian et al., 2012*. Results revealed that mean ranges vary between individual cancer types, and that $\mu$ shows more consistent significant difference between response categories relative to $\Lambda$ (these results are consistent with the results shown in *Butner et al., 2020*; these results are shown in *Appendix 1—figure 3*, *Appendix 1—figure 3—source data 1*, and explored further in 'Discussion').

## Confirmation of model stability

The parameters $\alpha$, $\Lambda$, and $\mu$ were held constant during the sensitivity analysis at three different values for each parameter. These values were the minimum, average, and maximum values derived from fitting *Equation (1)* to the data as described above. Changing $\Lambda$ and $\mu$ from these values ± 10% yielded a maximum change in $\rho'$ of 9.2% at $t$ = 200 days. Varying $\rho'$ data points randomly up to ±0.1 resulted in Spearman correlation coefficients of 0.766 and 0.919 for $\Lambda$ and $\mu$, respectively, derived from fitting these randomized data versus the actual clinical data. We note that, due to high stability observed to these perturbations in the large calibration cohort (n = 189 patients), we did not repeat this sensitivity analysis in the validation cohort. $\Lambda$ values derived from fitting truncated $\rho'$ data for the calibration cohort at various time points displayed Spearman correlation coefficients with $\Lambda$ derived from fitting the full data ranging from 0.071 at $t$ < 30 days to 0.730 at $t$ < 200 days, while Spearman correlation coefficients for $\mu$ ranged from 0.910 ($t$ < 30 days) to 0.921 ($t$ < 200 days) for all truncated datasets (*Table 2*). In the validation cohort, Spearman correlation coefficients for $\Lambda$ ranged 0.800 at $t$ < 30 days to 0.771 at $t$ < 200 days, and 0.800 at $t$ < 30 days to 0.989 at $t$ < 200 days for $\mu$. Simulated RECIST v1.1 categorization (complete/partial response vs. stable/progressive disease) by using the values of $\Lambda$ and $\mu$ derived from fitting the truncated calibration cohort data resulted in 19% misclassification by using data from $t$ < 60 days (n = 177) and 13% misclassification at $t$ < 200 days (n = 189) when compared to the full dataset. In the validation cohort, we observed 18.6% misclassification at $t$ < 60 days (n = 43) and 10.9% misclassification at $t$ <200 days (n = 64). Predicted normalized tumor mass after initiation of immunotherapy is depicted at different combinations of $\alpha$, $\Lambda$, and $\mu$ (*Figure 4*).

## Analysis of model predictions

Using the binary classification of tumor response of partial/complete response as positive and stable/progressive disease as negative, receiver-operator characteristic (ROC) analysis was performed to identify optimal predictive response thresholds for $\mu$ and $\Lambda$ (*Figure 3*, insets) as higher values of $\mu$ and $\Lambda$ are expected to correspond to a more favorable patient response. Identification of optimal predictive thresholds in the calibration cohort by maximizing the Youden's J statistic revealed cutoff thresholds where sensitivity (the proportion of correctly identified patients with partial/complete response) was higher for both $\Lambda$ and $\mu$ (0.945 and 0.891, respectively), while specificity was reduced (0.381 for $\Lambda$ and 0.567 $\mu$). Response prediction thresholds were identified as 0.722 for $\Lambda$ and 0.00905 for $\mu$. Testing these same response threshold values (identified in the calibration cohort) in the validation cohort revealed sensitivity of 0.6 for $\Lambda$ and for 0.960 $\mu$, and specificity of 0.743 for $\Lambda$ and 0.769 for $\mu$ in the validation cohort. We found that, in the calibration cohort, positive predictive values (PPVs; the probability a patient will be a responder when the model predicts they will be a responder: 0.381 for

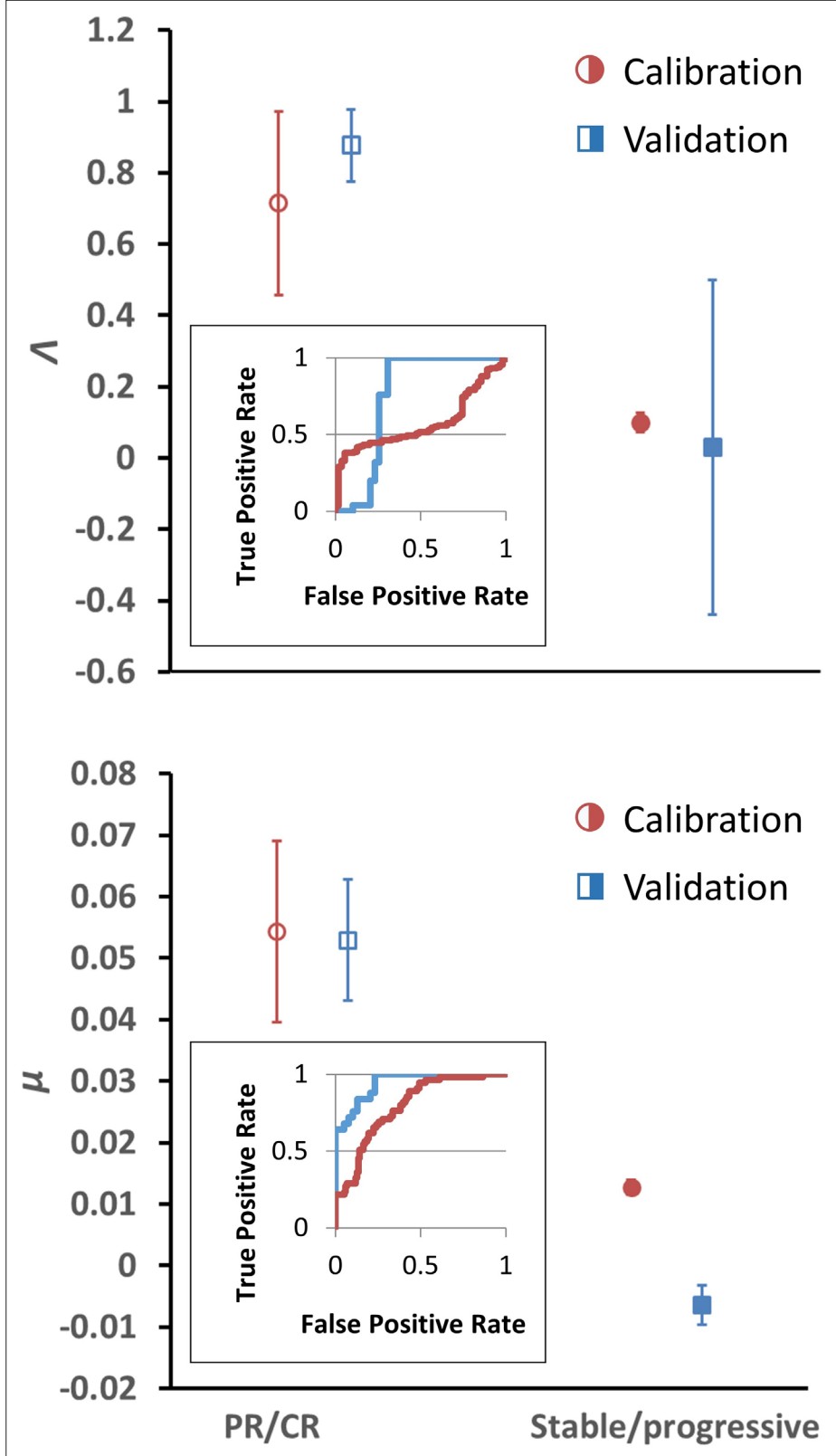

**Figure 3.** Depiction of average $\Lambda$ and $\mu$ values in patients with response (n = 55) versus nonresponse (n = 134) in the calibration cohort (circular markers), while n = 25 patients had objective response and 39 patients demonstrated stable/progressive disease in the validation cohort (square markers) as determined by RECIST v1.1 criteria. Open markers represent the average values of patients with response, and solid markers represent

*Figure 3 continued on next page*

*Figure 3 continued*

patients with stable/progressive disease. Error bars represent the standard error of the mean (SEM). p-Values of separation between groups by Wilcoxon rank sum (two tails): $\Lambda$, p=0.119 and p<0.001 for literature (calibration) and non-small cell lung cancer (NSCLC) (validation) cohorts, respectively; $\mu$, p<0.001 for both literature (calibration) and NSCLC (validation) cohorts. Insets: receiver-operator characteristic (ROC) curves for patient response versus model parameters for both cohorts; $\Lambda$, literature cohort: sensitivity = 0.381, specificity = 0.945, accuracy = 545; $\mu$, literature cohort: sensitivity = 0.891, specificity = 0.567, accuracy = 0.661; $\Lambda$, NSCLC clinical cohort: sensitivity = 0.600, specificity = 0.744, accuracy = 0.688; $\mu$, NSCLC clinical cohort: sensitivity = 0.960, specificity = 0.769, accuracy = 0.844. PR, partial response; CR, complete response. Examples of cancer drug-specific parameter values may be found in *Butner et al., 2020*.

The online version of this article includes the following source data for figure 3:

**Source data 1.** Numerical data for *Figure 3*.

---

$\Lambda$ and 0.458 for $\mu$) were lower than the negative predictive values (NPVs; 0.889 for $\Lambda$ and 0.927 for $\mu$), indicating fewer false-negatives (stable/progressive disease) than false-positives (partial/complete response). This trend was confirmed in the validation cohort (a PPV of 0.600 for $\Lambda$ and 0.723 for $\mu$; an NPV of 0.744 for $\Lambda$ and 0.968 for $\mu$). Lastly, we note that overall accuracy was lower by both $\Lambda$ and $\mu$ in the calibration cohort than were observed in the validation cohort; this is presumably due to the misbalance of patients with ORR (29% = 55 patients) versus patients with stable/progressive disease (71%, = 134 patients) while sensitivity (correctly identified patients with ORR) was higher than specificity.

## Comparison of $\Lambda$ and $\mu$ with clinical and pathological data

Conversion of $\Lambda$ to an estimated intratumoral CD8+ T cell count yielded a predicted mean ± SEM of 3970 ± 1429 cells/mm² in patients with partial/complete response to checkpoint inhibition and 553 ± 147 cells/mm² in patients with stable/progressive disease (*Figure 5*) in the calibration cohort, and 4871 ± 567 cells/mm² versus 165 ± 2,604 cells/mm² in patients with partial/complete response versus stable/progressive disease in the validation cohort. Intratumoral CD8+ T cell counts referenced from *Tumeh et al., 2014* are derived from two scenarios, either encompassing data in the pretreatment setting only (n = 46) or data including pathological CD8+ T cell counts encompassing both pretreatment and on treatment (n = 23). In the pretreatment setting, the average intratumoral CD8+ T cell count was 2632 ± 518 cells/mm² in patients with partial/complete response to checkpoint inhibitors and 322 ± 133 cells/mm² in patients demonstrating stable/progressive disease, while including

---

**Table 2.** Spearman correlation coefficients between $\Lambda$ and $\mu$ derived from fitting truncated datasets versus full dataset.

*t*: days. Note that values of 1.000 are due to only a small number of patients (n = 4) that were imaged before *t* = 30 days in the validation cohort; these either did not have lesion volumes reassessed before the next reported time threshold (*t* = 60 days) or did not observe a change in monotonic relationships within this timeframe (*t* = 30–120 days).

| | Calibration cohort | | | | | Validation cohort | | | |
|---|---|---|---|---|---|---|---|---|---|
| $\Lambda$ | *t* < 60 | *t* < 120 | *t* < 200 | *t* = all days | $\Lambda$ | *t* < 60 | *t* < 120 | *t* < 200 | *t* = all days |
| *t* < 30 | 0.476 | 0.162 | 0.080 | 0.071 | *t* < 30 | 1.000 | 1.000 | 0.800 | 0.800 |
| *t* < 60 | | 0.416 | 0.309 | 0.306 | *t* < 60 | | 0.812 | 0.658 | 0.823 |
| *t* < 120 | | | 0.668 | 0.599 | *t* < 120 | | | 0.676 | 0.750 |
| *t* < 200 | | | | 0.730 | *t* < 200 | | | | 0.771 |
| $\mu$ | *t* < 60 | *t* < 120 | *t* < 200 | *t* = all days | $\mu$ | *t* < 60 | *t* < 120 | *t* < 200 | *t* = all days |
| *t* < 30 | 0.942 | 0.928 | 0.922 | 0.910 | *t* < 30 | 1.000 | 0.800 | 0.800 | 0.800 |
| *t* < 60 | | 0.968 | 0.941 | 0.946 | *t* < 60 | | 0.974 | 0.960 | 0.963 |
| *t* < 120 | | | 0.946 | 0.922 | *t* < 120 | | | 0.971 | 0.961 |
| *t* < 200 | | | | 0.921 | *t* < 200 | | | | 0.989 |

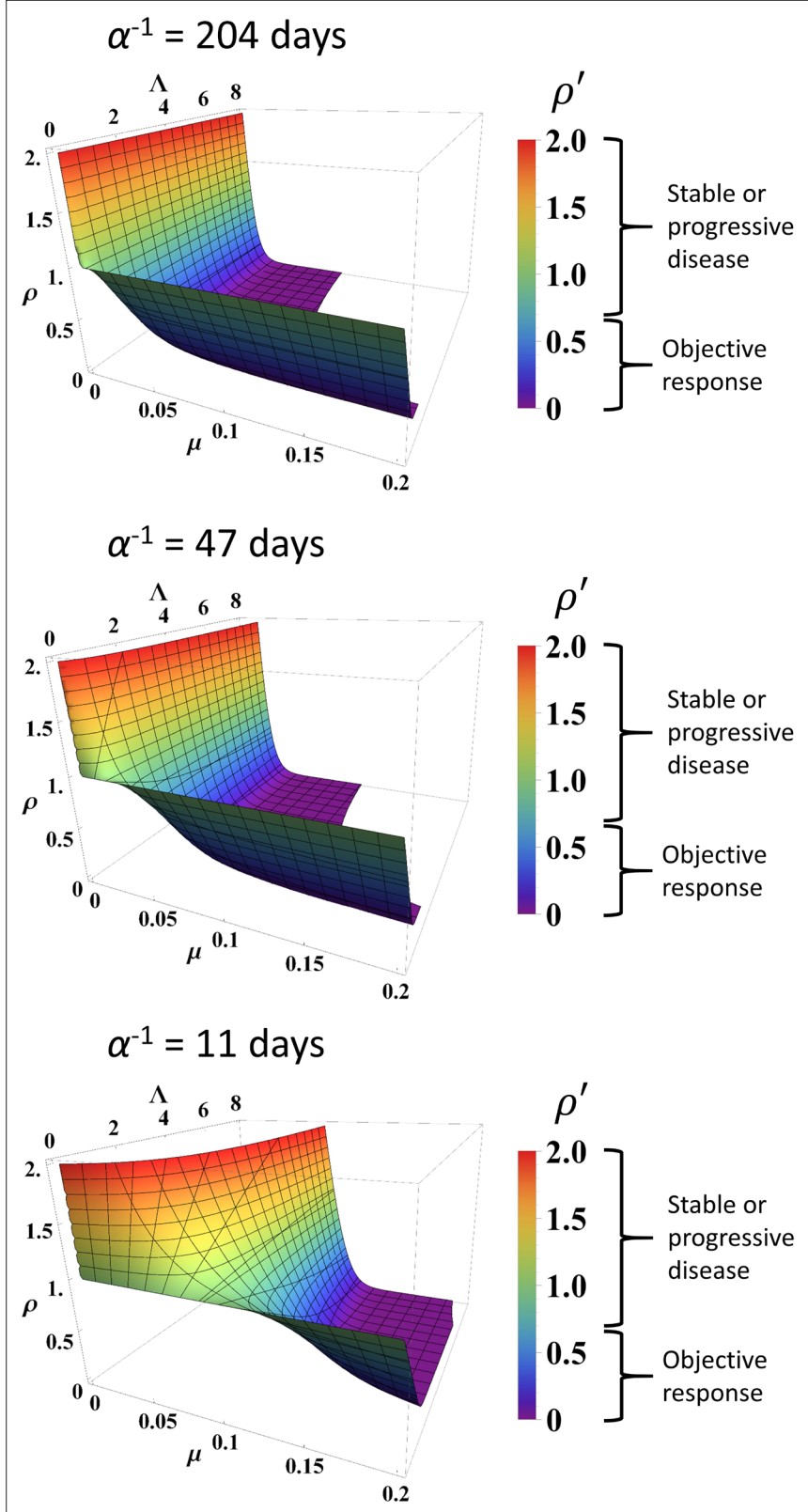

**Figure 4.** Simulated response to immune checkpoint inhibition at different values of α, Λ, and μ. Data are obtained from *Equation (1)*. Normalized tumor volume ($\rho'$) was determined at $t$ = 200 days. Three different α values were used that represent the minimum, average, and maximum values derived from fitting the calibration cohort, as described in the text. Λ and μ were varied continuously over their respective ranges. Colors also correspond with $\rho'$ as per color map on the right. RECIST v1.1 criteria of response are listed to the right of the color bars.

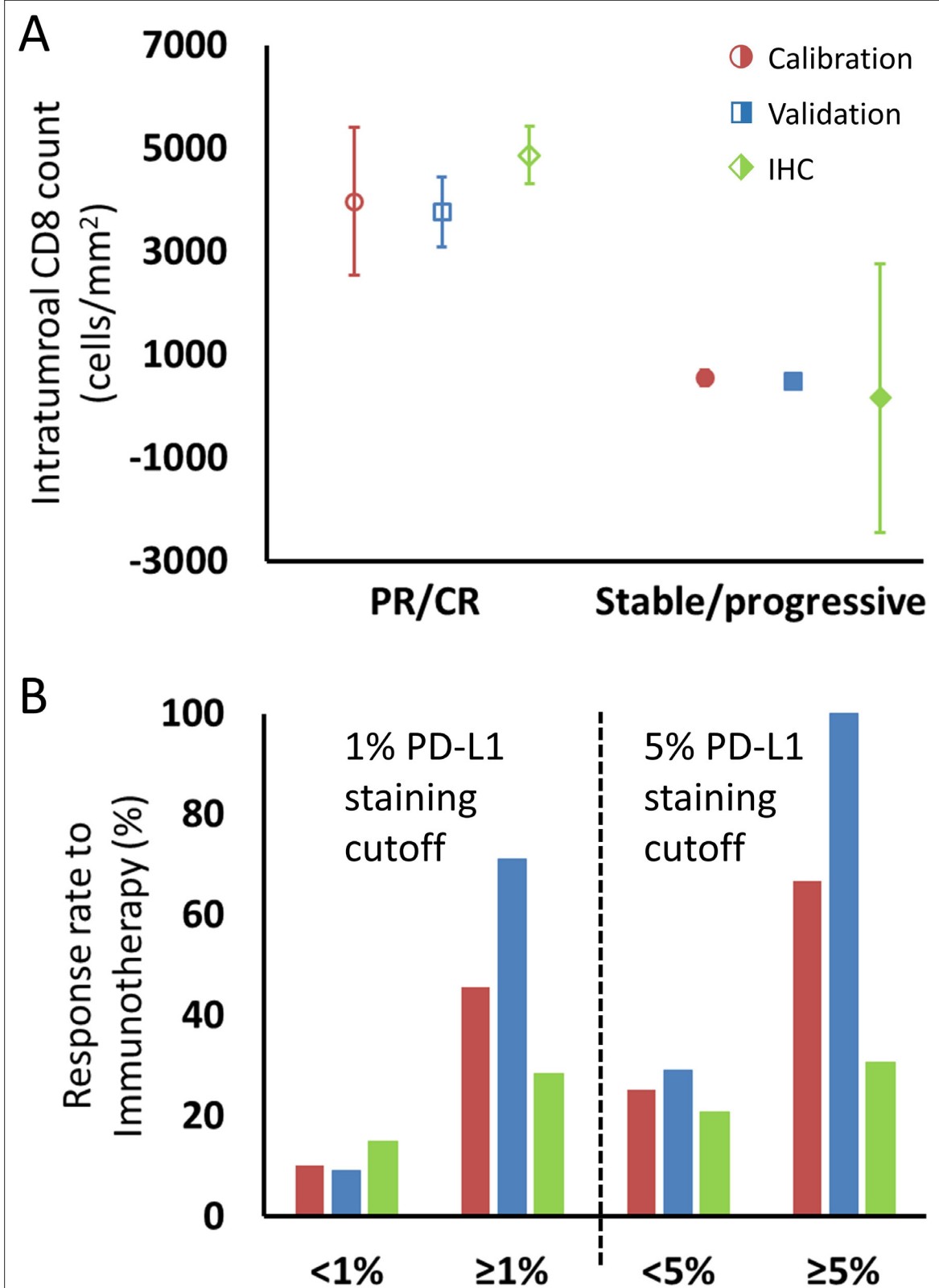

**Figure 5.** Comparison of intratumoral CD8+ T cell count and tumor PD-L1 staining derived from fitting the model to clinical data and values reported in the literature, as described in the text. (**A**) Model intratumoral CD8+ T cell count (circles: calibration cohort, p=0.119 [Wilcoxon, two-tail]; squares: validation cohort, p<0.001) was derived from Λ and literature CD8 intratumoral count was taken from immunohistochemical (IHC) staining in *Tumeh et al., 2014* in melanoma (diamonds; average CD8 counts including on-treatment values [n = 23]). CD8+ T cell counts from pretreatment biopsies

*Figure 5 continued on next page*

*Figure 5 continued*

only (n = 46) demonstrated mean values (± SEM) of 2632 ± 518 cells/mm² in patients with response to immunotherapy and 322 ± 133 cells/mm² in nonresponding patients, respectively. Values for CD8+ T cell counts are plotted as averages with error bars representing the standard error. (**B**) Patient response rates to immunotherapy stratified by PD-L1 staining were derived from $\mu$ from the model (calibration: red; validation: blue) and from references (*Borghaei et al., 2015*; *Robert et al., 2015b*; *Brahmer et al., 2012*; *Tumeh et al., 2014*; *Motzer et al., 2015*; *Powles et al., 2014*; *Topalian et al., 2012*; *Garon et al., 2015*; *Herbst et al., 2014*; *Kefford et al., 2014*; *Spira et al., 2015*; *Taube et al., 2014*; *Weber et al., 2015*) for the literature data (green; n = 975 for 1% cutoff, n = 1492 for 5% cutoff; see Appendix 1—table 1). Response to immune checkpoint inhibition was determined by RECIST v1.1 criteria. PR, partial response; CR, complete response.

The online version of this article includes the following source data for figure 5:

**Source data 1.** Numerical data for *Figure 5*.

on-treatment plus pretreatment data yielded an average of 3770 ± 675 cells/mm² in patients with partial/complete response and 501 ± 113 cells/mm² in patients with stable/progressive disease.

For comparison of $\mu$ with PD-L1 staining, data were extracted from a total of 12 published clinical trials (n = 975 cancer patients) with data on clinical response by using a PD-L1 staining cutoff of ≥1% versus <1% and 1492 patients with a cutoff of ≥5% versus <5%. The ORR of patients with these thresholds was 15% for PD-L1 staining <1% and 28% for PD-L1 staining ≥1%, and 21% and 38% for <5% and ≥5%, respectively. The ORR of patients in the calibration cohort by using the same thresholds in the derived values of $\mu$ was therefore 10% and 45% for patients with $\mu < 1\%$ and ≥1% , respectively, and was 25% and 67% for patients with $\mu < 5\%$ and ≥5%, while in the validation cohort, the model-predicted ORR was 9% and 71% for patients with $\mu < 1\%$ and ≥1% , respectively, and was 29% and 100% for patients with $\mu < 5\%$ and ≥5% (*Figure 5—source data 1*).

## Discussion

The translational mathematical model introduced here retrospectively investigates the molecular, cellular, and biophysical mechanism(s) behind patient response to immune checkpoint inhibitor therapy, and it explores the potential clinical value of the early prediction of treatment effectiveness. The reductionist approach introduced here simplifies the rather complex biological cross-talk between the immune system with cancer cells to a set of differential equations ultimately dependent on three intrinsic parameters that represent tumor proliferation, intratumoral immune cell presence/killing efficiency (considered as baseline immune cell infiltration), and checkpoint inhibition efficacy; this latter parameter represents the active role of the checkpoint inhibitors in initiating and amplifying the anti-tumor response. We have demonstrated that these parameters may be informed using either imaging or IHC measures, enabling for robust implementation even when some of these measures are unavailable, and offering methods to calculate PD-L1 density and intratumoral CD8+ T cell counts without invasive methods or to inform the model using only IHC measures at start of treatment.

From a mechanistic standpoint, the mathematical model is fully compatible with the current understanding of the biological effect of checkpoint blockade and clinical subtypes of immunotherapy-responsive tumors (*Teng et al., 2015*). First, the parameter values derived from our model for intratumoral immune cell infiltration ($\Lambda$) and immunotherapy antibody efficacy ($\mu$) correspond with the quantitative pathological measures of intratumoral CD8+ T cell infiltration and PD-L1-positive staining (*Figure 4*; *Figure 5*). Although parameters $\Lambda$ and $\mu$ are not direct representations of CD8+ T cell counts or PD-L1 expression, they correlated with measured pathological biomarkers, thus providing strong experimental evidence that these mathematical parameters may be quantified from these biomarkers. This offers a unique method for clinical translation using clinically available measured quantities to provide personalized prediction of patient response, a potential that we have often found particularly lacking in prior mechanistic modeling research. By demonstrating links between model parameters and known biology, our model may also be used to quantify the associated mechanistic underpinnings of treatment failure, with the goal of suggesting clinical interventions that may overcome these deficiencies to improve patient outcome. Unfortunately, the absence of per-patient measurements of CD8+ T cell counts or PD-L1 expression corresponding to individual patient response to treatment in the cohorts examined prevented analysis of how these parameters perform on a per-patient basis. We are currently collecting in-house a new dataset containing both measures for single patients, and the results of this study will be published once complete. We are also investigating how these biomarkers

may inform the model to prospectively identify pseudoprogression (a clinical phenomenon wherein lesions are observed to experience rapid volume increase followed by disease response) on a per-patient basis, based on our previous result that retrospectively known long-term parameter values are unique in pseudoprogression (**Butner et al., 2020**).

Moreover, the model demonstrates that the rate of immunotherapy response arises from nonlinear interactions between both adequate effector immune cell infiltration in tumors and immunotherapy efficacy within the tumor microenvironment (**Figure 4**). Mechanistic mathematical modeling across several cancer types (**Table 1**) suggests that the overlying master equation [i.e., **Equation (1)**] may well be universally applied, while the model parameter values presented here are likely tumor-specific. Indeed, this conclusion may be observed in **Figure 3**, where $\Lambda$ and $\mu$ are distinctly different for ORR versus stable or progressive disease, but the values are different for the different cohorts, thereby capturing mechanistic differences across the different cancer types. However, further testing of the model in additional cancer types and larger patient cohorts remains necessary, in part due to the heterogeneous nature of solid tumors, both within individual patients and across different cancer types. The nontrivial interaction of these inputs predicted by this model is different in comparison to many established and current candidate biomarker studies (which focus on linear immunotherapy responses to just one of these markers, e.g., PD-L1 expression [**Carbognin et al., 2015**]), perhaps providing a plausible explanation why single biomarkers sometimes provide inconsistent predictions across multiple cancer types (**Carbognin et al., 2015**). Similarly, the unique insights presented herein allow for a more direct comparison of the model parameters to common pathological markers either reported in the literature or as yet to be reported, and provide a different perspective compared to other immune checkpoint inhibitor modeling studies (**Łuksza et al., 2017**; **Wilkie and Hahnfeldt, 2013**; **Serre et al., 2016**).

The high sensitivity at early time points ($t < 60$ days) demonstrated here indicates that this model may provide valuable early identification of cancer patients most likely to benefit from checkpoint inhibition therapy, with a high NPV indicating minimal false-negatives in the group predicted to receive the least benefit from therapy. Therefore, this attribute could serve as an early indicator of the efficacy of checkpoint inhibition by simply using either clinical or imaging measurements of index tumor lesions and by applying the mathematical model to project maximal effect for an individual patient. Further, the model demonstrates robustness and stability in the derived values of $\Lambda$ and $\mu$ to random variations in $\rho'$, and by using values of $\Lambda$ and $\mu$ derived from fitting limited data over time (**Table 2**). This result is in stark contrast to previous mathematical descriptions of immune-mediated killing of cancer cells, where a relatively small change in a single input parameter of merely ±1% could dramatically vary tumor response by up to ±50% (**de Pillis et al., 2005**).

Towards clinical translation of cancer models, a mathematical model with two simple terms (an exponential growth term and a regression term) has previously been used to successfully correlate and predict patient survival (**Stein et al., 2008**). The fundamental difference between this prior model (**Stein et al., 2008**) and the model reported here is that the quantification of cell death is more mechanistic, that is, depending on the specific mechanisms (**Figure 1**) that underlie cancer cell and immunotherapy drug interactions, hence allowing one to gain more mechanistic insight into the treatment system. As demonstrated, model parameters $\Lambda$ and $\mu$ can be used to stratify responding and nonresponding patients (**Figure 3**), which has immediate potential utility as a biomarker for patient selection in prospective clinical trials. Moreover, the model contains biological values that, at least in principle, may either be measured (e.g., tumor burden, growth rate) or changed clinically ($\mu$: e.g., drug dosing or dosing schedule; $\Lambda$: e.g., radiotherapy-induced increase in PD-L1 expression, radiotherapy abscopal effect). Thus, we expect that the model (i) may be informed by using standard-of-care measurements (e.g., **Figure 5**) and (ii) can provide quantitative information on which parameters must be changed to maximize therapeutic benefit, providing clinicians with potentially valuable decision-making information.

Of course, as with any biomathematical model, this model is based on several educated assumptions about the interactions of immune effectors (CD8+ T cells) with cancer cells, which naturally does not reflect the sheer magnitude and complexity of the tumor-immune microenvironment. These include the presence of immunosuppressive cells (such as neutrophils, myeloid-derived suppressor cells, Treg, and M2-polarized tumor-associated macrophages), which extensively infiltrate into tumor mass and hinder the efficacy of immune checkpoint inhibitors (**Jenkins et al., 2018**). Moreover, effects

of other immune cells, such as CD4+ T cells, are not explicitly included in the current model form; while it is likely that some of the effects from these other cells are captured within the model representation of CD8+ T cells, this assumption warrants future model development where each immune component is explicitly represented. Some of these assumptions (e.g., lack of immune cell/antibody infiltration terms) are supported with in vitro or in vivo data such as high rates of immune cell binding by anti-PD1 antibodies in the peripheral circulation (*Topalian et al., 2012*) and local intratumoral (as opposed to systemic myelogenous) expansion of CD8+ T cells during checkpoint inhibition (*Tumeh et al., 2014*; *Ribas et al., 2016*). Other potential model parameters would logistically be quite challenging to measure in an outpatient clinic-ready setting routinely at this time, and their inclusion remains open for follow-up confirmatory retrospective studies and ultimately for future prospective clinical trials. As such, the integration of additional prognostic biomarkers and master regulators that intrinsically (e.g., genetic stability) and extrinsically control the efficacy of immunotherapy represents open research areas for future translational investigation.

Moreover, the two response parameters derived from the model, $\Lambda$ and $\mu$, represent abstract terms, which may not precisely model intratumoral CD8+ lymphocyte infiltration and tumor cell PD-L1 staining and potentially can be modified by relevant determinants, such as tumor mutational burden (*Goodman et al., 2017*) or the human microbiome of the gut (*Gopalakrishnan et al., 2018*). Moreover, these parameters represent a simplifying average value of these fluctuating parameters. We take this to be a reasonable assumption because (i) these quantities are not directly measurable (this is also the case for cell proliferation events underlying the tumor growth rate parameter $\alpha$), (ii) in our in-house data, times between patient reassessment ranged 17–91 days, so even direct measurement of these would provide an average value over this time period, and (iii) these processes take place at far shorter times than patient reassessments. By presenting the model in this reduced form, we enable simplified interpretation of the results while also providing a single, easy-to-understand scalar that contains significant information about the treatment response.

Finally, a few operational aspects of this work deserve further comment. Response was defined using RECIST v1.1 criteria, *in lieu* of the more recent adaptation of RECIST criteria to immunotherapy treatment (iRECIST), because the studies used to derive the model parameters in this mechanistic analysis did not incorporate these new criteria, and our use of published data for the calibration cohort limited or precluded us from obtaining a priori knowledge about nontarget lesions. Moreover, our work to date has only involved testing our model in a handful of cancer types, and further validation in additional disease phenotypes and larger total patient populations remains outstanding. These inherent study limitations notwithstanding, we will continue to improve our model to more accurately predict outcomes in the immunotherapy-specific setting of individualized cancer patients, potentially through the inclusion of additional biomarkers, as bioactive molecules such as IFNγ, CD206, CX3CR1, CD1D, and iNOS, along with cell-mediated mechanisms known to have an effect on immune checkpoint inhibitor therapy (*Park et al., 2018*; *Gubin et al., 2018*). However, we do not anticipate that such improvements and refinements would change the basic findings and the potential value of the mechanistic mathematical model reported in this work. In going forward, one hopes that knowledge broadly applicable across multiple cancer types and/or immune microenvironment control will be serially incorporated to the dynamic model upon their future discovery and validation.

In conclusion, we present a mechanistic model of immune checkpoint inhibition that is able to describe various immunotherapy response profiles, with the inputs to the model correlating with common pathological biomarkers in current clinical use. An early and robust a priori predictor for checkpoint inhibition response and outcome might provide a glimpse of the immense potential for timely adjustments and therapeutic personalization. Future prospective investigations of this computational science-assisted approach will focus on readily available clinical data as inputs to the model and further refining the complex interplay between the immune system and the cancer environment to extract other important variables for immunotherapy efficacy. Towards this end, we are currently pursuing in-house collection of per-patient, paired IHC measures of PD-L1 or T cell counts with tumor response, which will enable us to directly correlate IHC measures of interest with tumor response, while also allowing for examination of additional patient parameters, such as tumor stage or patient age. We will also obtain data from different cancer types known to be receptive to ICI therapy in order to further evaluate the potential of our model to perform as a more universal predictor across an even more diverse array of cancer phenotypes. Ultimately, the merit of this approach will rely on its future

ability to reliably predict early individual patient response with the goal of improving personalized cancer care.

## Additional information

### Competing interests

David S Hong, James W Welsh, Elizabeth A Mittendorf, Shridar Ganesan: See COI form submitted. The other authors declare that no competing interests exist.

### Funding

| Funder | Grant reference number | Author |
|---|---|---|
| National Science Foundation | DMS-1930583 | Zhihui Wang<br>Vittorio Cristini |
| National Institutes of Health | 1R01CA253865 | Zhihui Wang<br>Vittorio Cristini |
| National Institutes of Health | 1U01CA196403 | Zhihui Wang<br>Eugene J Koay<br>Vittorio Cristini |
| National Institutes of Health | 1U01CA213759 | Zhihui Wang<br>Vittorio Cristini |
| National Institutes of Health | 1R01CA226537 | Zhihui Wang<br>Renata Pasqualini<br>Wadih Arap<br>Vittorio Cristini |
| National Institutes of Health | 1R01CA222007 | Zhihui Wang<br>Vittorio Cristini |
| National Institutes of Health | U54CA210181 | Zhihui Wang<br>Mauro Ferrari<br>Shu-Hsia Chen<br>Ping-Ying Pan<br>Eugene J Koay<br>Vittorio Cristini |
| National Institutes of Health | P30CA016672 | David S Hong<br>James W Welsh<br>Eugene J Koay |
| National Institutes of Health | P30CA072720 | Steven K Libutti<br>Shridar Ganesan<br>Renata Pasqualini<br>Wadih Arap |
| European Commission | SER Cymru II Programme | Bruna Corradetti |
| National Institutes of Health | U01CA200468 | Eugene J Koay |
| National Institutes of Health | R01CA222959 | Mauro Ferrari |
| DOD Breast Cancer Research | Breakthrough Level IV Award W81XWH-17-1-0389 | Mauro Ferrari |
| AngelWorks | | Renata Pasqualini<br>Wadih Arap |
| Gillson Longenbaugh Foundation | | Renata Pasqualini<br>Wadih Arap |
| Marcus Foundation | | Renata Pasqualini<br>Wadih Arap |
| National Institutes of Health | R01CA109322 | Shu-Hsia Chen |

| Funder | Grant reference number | Author |
| --- | --- | --- |
| National Institutes of Health | R01CA127483 | Shu-Hsia Chen |
| National Institutes of Health | R01CA208703 | Shu-Hsia Chen |
| National Institutes of Health | R01CA140243 | Ping-Ying Pan |
| National Institutes of Health | R01CA188610 | Ping-Ying Pan |

The funders had no role in study design, data collection and interpretation, or the decision to submit the work for publication.

## Author contributions

Joseph D Butner, Data curation, Formal analysis, Investigation, Methodology, Resources, Software, Validation, Writing – original draft, Writing – review and editing; Geoffrey V Martin, Data curation, Formal analysis, Investigation, Methodology, Resources, Software, Writing – original draft, Writing – review and editing; Zhihui Wang, Formal analysis, Funding acquisition, Investigation, Methodology, Project administration, Resources, Supervision, Writing – original draft, Writing – review and editing; Bruna Corradetti, Mauro Ferrari, Nestor Esnaola, Caroline Chung, David S Hong, Naomi Hasegawa, Elizabeth A Mittendorf, Steven A Curley, Shu-Hsia Chen, Ping-Ying Pan, Steven K Libutti, Shridar Ganesan, Richard L Sidman, Investigation, Writing – review and editing; James W Welsh, Data curation, Investigation, Resources, Writing – review and editing; Renata Pasqualini, Wadih Arap, Investigation, Methodology, Resources, Writing – review and editing; Eugene J Koay, Conceptualization, Formal analysis, Funding acquisition, Investigation, Methodology, Project administration, Resources, Supervision, Writing – original draft, Writing – review and editing, Data curation; Vittorio Cristini, Conceptualization, Formal analysis, Funding acquisition, Investigation, Methodology, Project administration, Resources, Supervision, Writing – original draft, Writing – review and editing

## Author ORCIDs

Joseph D Butner http://orcid.org/0000-0003-0608-2580
Zhihui Wang http://orcid.org/0000-0001-6262-700X
Wadih Arap http://orcid.org/0000-0002-8686-4584
Eugene J Koay http://orcid.org/0000-0001-7675-3461

## Ethics

Clinical trial registration NCT02444741.
Human subjects: In-house patient cohort were obtained as de-identified data from a study conducted in accordance with the U.S. Common Rule and with Institutional Review Board Approval at MD Anderson (2014-1020), including waiver of informed consent. This work has been published in J Immunother Cancer. 2020; 8(2): e001001. PMC7555111. doi: 10.1136/jitc-2020-001001.

## Decision letter and Author response

Decision letter https://doi.org/10.7554/eLife.70130.sa1
Author response https://doi.org/10.7554/eLife.70130.sa2

# Additional files

## Supplementary files

• Appendix 1—figure 3—source data 1. Numerical data for *Appendix 1—figure 3*.
• Transparent reporting form

## Data availability

No new clinical patient data was produced in this study. Data used that was obtained from literature is available in the original publications; we have been careful to cite each of these in the manuscript. Interested researchers should reach out directly to the primary authors of these studies. Data for

the in-house clinical trial cohort are from the study reported in PMC7555111; interested researchers should contact the authors of this publication with any data requests.

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

# Appendix 1

## Immunotherapy model derivation

The mechanistic model of cancer immunotherapy presented in this paper is essentially an extension of our prior mathematical model describing chemotherapy response that has been shown to predict cancer cell kill with various cytotoxic or targeted agents. In particular, we extend these equations to consider immune cell presence as necessary for cancer cell kill secondary to immunotherapy, and that immune cells respond to not only diffusion gradients within a tumor microenvironment but can also be influenced by the presence of cytokines through cell signaling pathways and chemotaxis. The model hypothesis of immunotherapy presented here is that these molecules or antibodies diffuse and block the complimentary interaction between immune checkpoint ligands expressed on cancer cells with their complementary proteins on immune cells. This mechanism renders effector immune cells potentially active for cancer cell killing; thus, the therapeutic site of action occurs within the tumor microenvironment (i.e., because our model only describes the tumor region and the factors and processes contained therein, we have made the assumption that all key mechanisms such as drug binding occur only within the tumor). These assumptions lead to the following system of PDEs:

$$\frac{\partial \rho}{\partial t} = \alpha \cdot \rho - \lambda_{\mathrm{p}} \cdot \rho \cdot \psi_k \cdot \int_0^t \lambda \cdot \sigma \cdot \rho \cdot \mathrm{d}t' \tag{A1}$$

$$\frac{\partial \psi_k}{\partial t} = -\chi \cdot \nabla \cdot \left( \psi_k \cdot \nabla \cdot C \right) + \Lambda_\psi \cdot \left( \frac{\partial \rho}{\partial t} - \alpha \cdot \rho \right) \tag{A2}$$

$$0 = D_a \cdot \nabla^2 \sigma - \lambda \cdot \sigma \cdot \rho \tag{A3}$$

$$0 = D_C \cdot \nabla^2 C + \Lambda_C \cdot \lambda \cdot \sigma \cdot \rho \tag{A4}$$

Here $\rho$, $\psi_k$, $\sigma$, and $C$ represent the local concentration of cancer cells, therapeutic immune cells, immunotherapy antibodies, and cytokines, respectively. Moreover, $\alpha$ is the proliferation rate of tumor cells, $\lambda_p$ is the specific death rate of cancer cells, $\lambda$ is the binding coefficient of immunotherapy antibodies that block the interaction between immune inhibitory ligands on cancer cells and their counterparts on immune cells, $\Lambda_\psi$ is the coupling of immune cell activity relative to the number of cancer cells (i.e., killing and tumor infiltration capacity of the host immune system), $\chi$ is the chemotaxis coefficient, $\Lambda_C$ is the number of released cytokines as immunotherapy antibodies are bound, and $D_a$ and $D_c$ are the diffusivity of antibodies and cytokines (see also *Figure 1*, main text).

The first equation describes the death rate of cancer cells as proportional to the concentration of therapeutic immune cells and the time history of immunotherapy antibody uptake and binding within the tumor environment. The second equation represents mass conservation of 'effective' therapeutic immune cells, including the rate at which these cells become ineffective at killing cancer cells. The third and fourth equations represent the concentration and diffusion of immunotherapy antibodies and cytokines in the presence of immunotherapy antibody binding.

To compare this mechanistic model with clinical immunotherapy data, we make multiple reductionist assumptions about the influence of various parameters from *Equation A1-4* on immunotherapy response. These assumptions suppose that immune cells are in relatively close physical proximity to cancer cells, allowing one to ignore the influence of cytokines to guide their movement, and that the primary effector immune cell response to immunotherapy is driven by the presence of immune cells already present in the tumor at the initiation of the therapy. Furthermore, we assume uptake and binding of immunotherapy antibodies within the tumor environment to occur diffusely relative to cancer and immune cell concentrations; therefore, in this model, we set $\chi = 0$ and remove the influence of immunotherapy antibody concentration. These assumptions reduced the four PDEs to the following set of coupled ODEs as follows:

$$\frac{d\rho}{dt} = \alpha \cdot \rho - \lambda_{\mathrm{p}} \cdot \rho \cdot \psi_k \cdot \int_0^t \lambda \cdot \sigma \cdot \rho \cdot \mathrm{d}t' \tag{A5}$$

$$\frac{d\psi_k}{dt} = \Lambda_\psi \cdot \left( \frac{d\rho}{dt} - \alpha \cdot \rho \right) \tag{A6}$$

Here, *Equation A6* represents the change in tumor-infiltrating immune cells relative to the change in cancer cells killed by the immune system. In the case where immune cell killing is weak (i.e., $\lambda_p \cong 0$, and thus $\frac{d\rho}{dt} = \alpha \cdot \rho$), the change in immune cell concentration within the tumor is negligible compared to the growth of the tumor, thus *Equation A6* satisfies the relationship $\frac{d\psi_k}{dt} \cong 0$ (i.e., immune cell concentration within the tumor remains roughly constant over time). When immune cell killing is sufficiently effective, then

$\lambda_{\mathrm{p}} \cdot \rho \cdot \psi_k \cdot \int_0^t \lambda \cdot \sigma \cdot \rho \cdot \mathrm{d}t' > \alpha \cdot \rho$ and thus $\frac{d\psi_k}{dt} \cong \Lambda_\psi \cdot \left( \frac{d\rho}{dt} \right)$. In essence, the change in immune cells in the tumor over time is roughly equal to the immune cell coupling to cancer cells [i.e., immune cells in the tumor environment are related to the tumor volume by a coupling factor ($\Lambda_\psi$)], which captures the immunogenicity of an individual tumor. Integrating both sides of *Equation A6* over time leads to the following relationship:

$$\psi_k - \psi_0 = \Lambda_\psi \cdot \left[ \rho - \rho_0 \right] \tag{A7}$$

where $\rho_0$ and $\psi_0$ represent the concentration of tumor cells and immune cells at the start of immunotherapy. We then substitute $\psi_k$ from, *Equation (A7) into Equation (A5)* express $\Lambda_{\psi\rho} = \Lambda_\psi \cdot \rho_0/\psi_0$ , and replace the concentration of tumor cells $\rho$ by a proportion of the original tumor cell concentration to obtain the normalized tumor mass $\rho' = \rho/\rho_0$ (while noting that one may also use normalized tumor volume here) to give us one equation of tumor response represented by

$$\frac{d\rho'}{dt} = \alpha \cdot \rho' - \lambda_{\mathrm{p}} \cdot \rho' \cdot \left[ 1 + \Lambda_{\psi\rho} \cdot \left( \rho' - 1 \right) \right] \cdot \int_0^t \lambda \cdot \sigma \cdot \rho' \cdot \mathrm{d}t' \tag{A8}$$

Finally, we assume that the binding of immunotherapy antibodies within the tumor environment ($\int_0^t \lambda \cdot \sigma \cdot \rho' \cdot \mathrm{d}t'$) and rate of tumor cell death secondary to effector immune cells ($\lambda_{\mathrm{p}}$) reach constant steady states on time scales faster than measurable tumor cell kill, implying $\lambda_{\mathrm{p}} \cdot \int_0^t \lambda \cdot \sigma \cdot \rho' \cdot \mathrm{d}t' = \mu$, leading to

$$\frac{d\rho'}{dt} = \alpha \cdot \rho' - \rho' \cdot \left[ 1 + \Lambda_{\psi\rho} \cdot \left( \rho' - 1 \right) \right] \cdot \mu \tag{A9}$$

This ODE can be solved analytically for $\rho'$ by integrating both sides, yielding

$$\rho' = \frac{1 + \frac{\alpha - \mu}{\mu \cdot \Lambda_{\psi\rho}}}{1 + \left( \frac{\alpha - \mu}{\mu \cdot \Lambda_{\psi\rho}} \right) e^{-\left( \alpha - \mu + \mu \cdot \Lambda_{\psi\rho} \right) \cdot t}} \tag{A10}$$

*Equation A10* is the same as *Equation (1)* in the primary text, which is used to compare with clinical response data to immunotherapy.

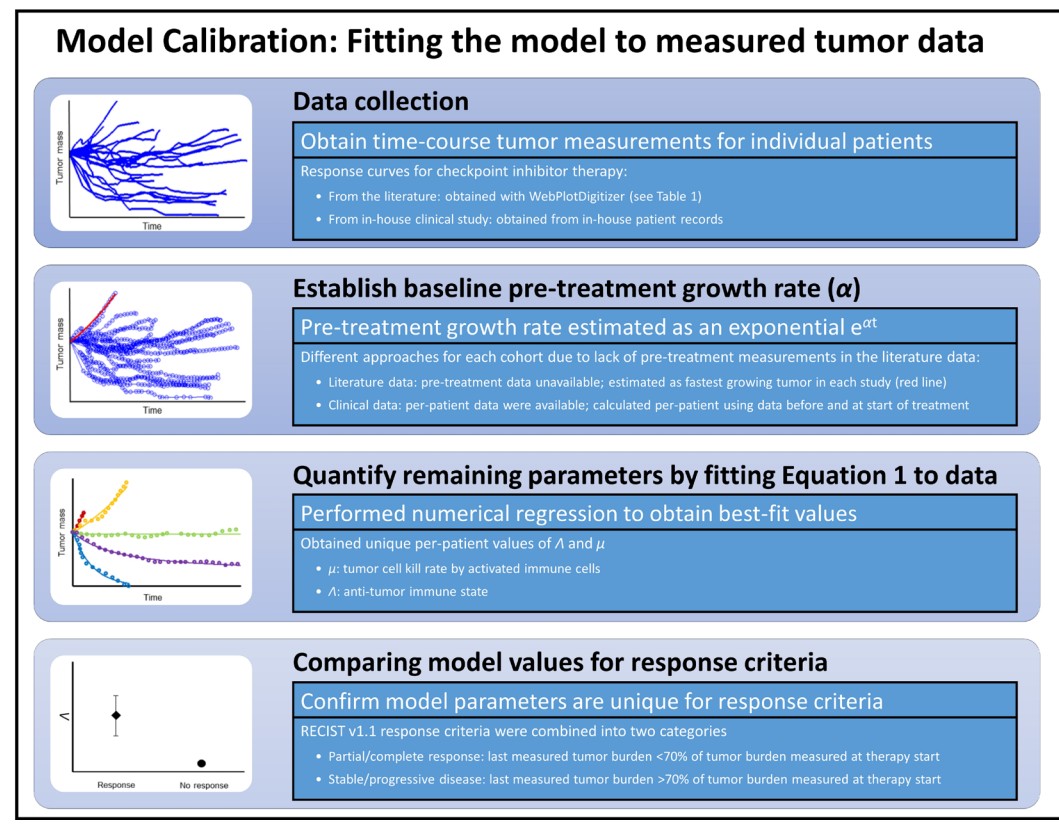

**Appendix 1—figure 1.** Steps for calibration of the mathematical model with clinical data. First, checkpoint inhibitor response curves were extracted from the literature. In all cases, immunotherapy treatment began at time $t = 0$. Second, a tumor-specific proliferation constant ($\alpha$) was determined for each cancer type by fitting exponential function ($e^{\alpha t}$ to fastest progressing patient in each clinical trial [red line]). Third, individual patient response data were fit to **Equation (1)** by using the respective $\alpha$ to determine $\Lambda$ and $\mu$. $\Lambda$ and $\mu$ values were then with compared in patients with partial/complete response versus patients with stable/progressive disease after immunotherapy by using the RECIST v1.1 criteria.

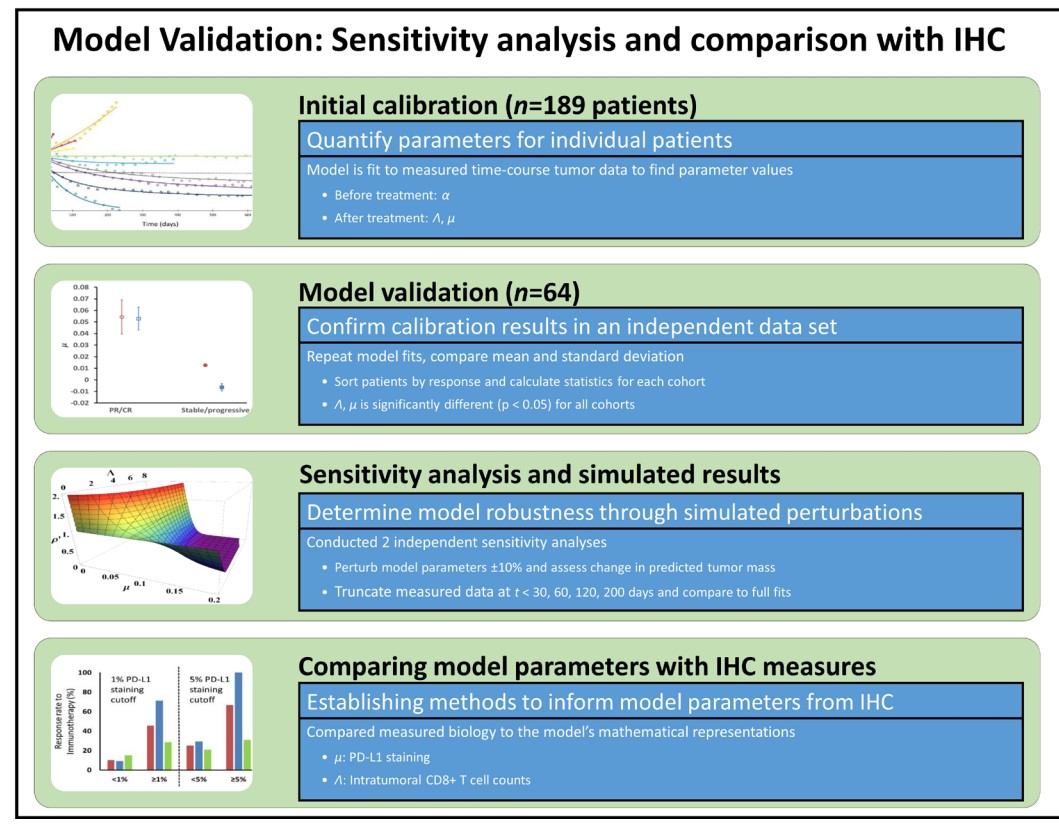

**Appendix 1—figure 2.** Model validation, sensitivity studies, and comparison of model parameters to immunohistochemical (IHC) measures. Model parameters were obtained from a second in-house patient cohort of patients with non-small cell lung cancer (NSCLC) (n = 64), which were compared to values obtained in the calibration cohort in a validation study. To study the sensitivity of the model to changes in model parameter values, key parameters were perturbed ±10% and the resultant simulated expected tumor burden was compared to measured values pre-perturbation. Tumor burden measures were also truncated, and results of truncated and full dataset model fits were compared. Lastly, the full parameter space of the model was examined. In order to compare model parameters to the underlying biology, model parameters were converted to intratumoral CD8+ lymphocyte counts (for $\Lambda$) and PD-L1 staining (for $\mu$), which were compared to IHC measures obtained from the literature.

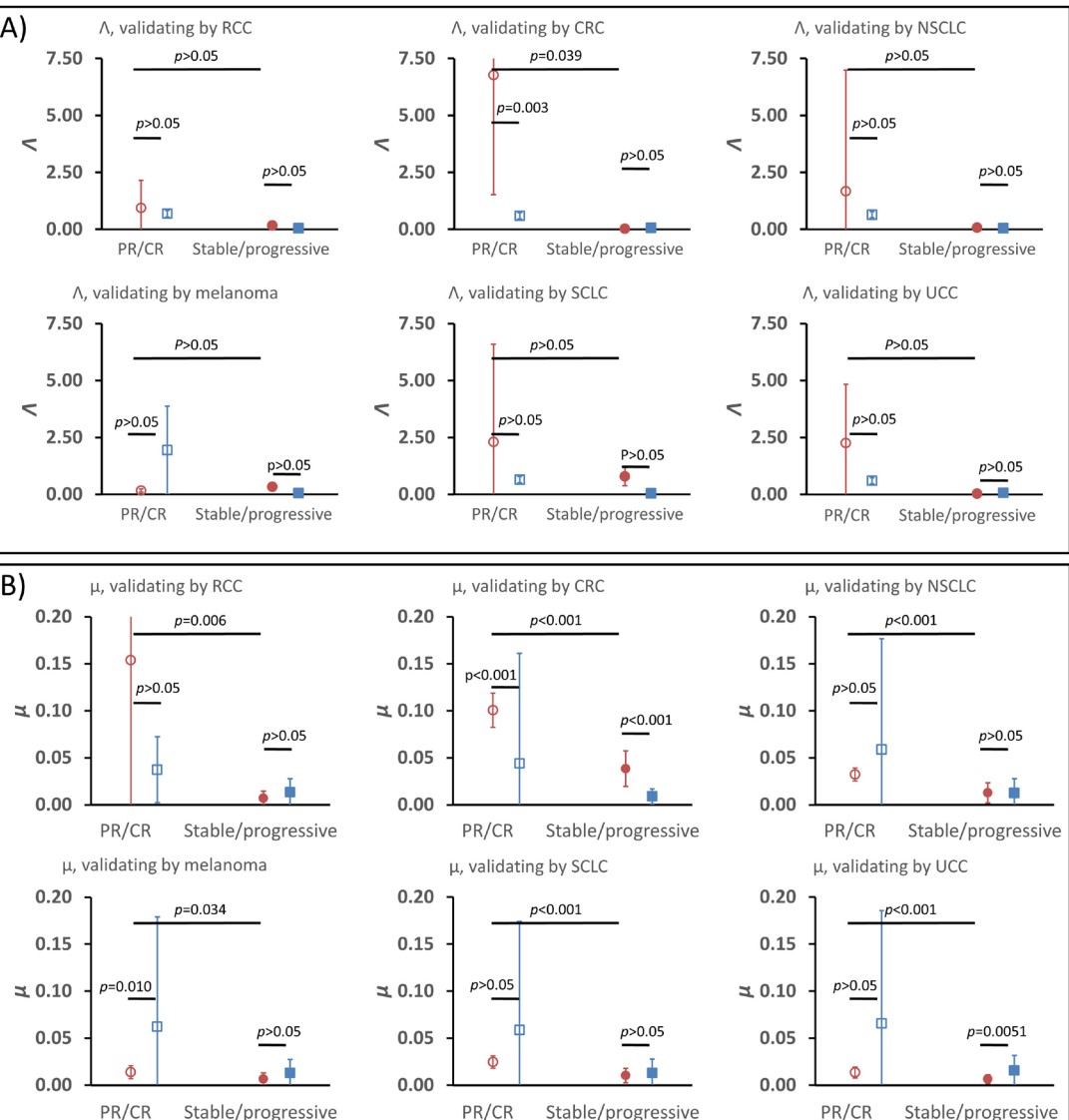

□ Calibration (PR/CR)   ■ Calibration (Stable/progressive)   ○ Validation (PR/CR)   ● Validation (Stable/progressive)

**Appendix 1—figure 3.** Parameter validation analysis within the calibration cohort. In order to examine the robustness of ranges for (**A**) parameter $\Lambda$ and (**B**) parameter $\mu$ between partial and complete response (PR/CR) versus stable/progressive disease among different cancer types, a validation study was performed where one cancer type was removed from the calibration cohort and used as validation against the parameter ranges in the reduced calibration set obtained from *Borghaei et al., 2015*; *Antonia et al., 2015*; *Le et al., 2015*; *Motzer et al., 2015*; *Powles et al., 2014*; and *Topalian et al., 2012*. Analysis was repeated once for each cancer type, and results are shown as mean ± standard deviation (error bars). Parameter ranges were found to vary between individual cancer types, and with $\mu$ showing more consistent significant difference between response categories relative to $\Lambda$ (these results are consistent with results shown in *Butner et al., 2020*).

The online version of this article includes the following source data for appendix 1—figure 3:

• **Appendix 1—figure 3—source data 1.** Numerical data for *Appendix 1—figure 3*.

**Appendix 1—table 1.** Studies used for derivation of pathological markers of immunotherapy response.

| Reference (see main text) | Tumor type | Checkpoint inhibitor | Pathological biomarker | PD-L1 staining cutoff |
|---|---|---|---|---|
| *Tumeh et al., 2014* | Melanoma | Pembrolizumab | CD8+ TILs | N/A |
| *Kefford et al., 2014* | Melanoma | Pembrolizumab | PD-L1 | 1% |
| *Powles et al., 2014* | UCC | Atezolizumab | PD-L1 | 1%, 5%, 10% |
| *Herbst et al., 2014* | NSCLC, RCC, melanoma, HNSCC, CRC, gastric and pancreatic cancer | Atezolizumab | PD-L1 | 1%, 5%, 10% |
| *Robert et al., 2015b* | Melanoma | Nivolumab | PD-L1 | 5% |
| *Motzer et al., 2015* | RCC | Nivolumab | PD-L1 | 5% |
| *Taube et al., 2014* | NSCLC, RCC, melanoma, PC, CRC | Nivolumab | PD-L1 | 5% |
| *Spira et al., 2015* | NSCLC | Atezolizumab | PD-L1 | 1%, 5%, 10% |
| *Brahmer et al., 2012* | NSCLC | Nivolumab | PD-L1 | 1%, 5%, 10% |
| *Borghaei et al., 2015* | NSCLC | Nivolumab | PD-L1 | 1%, 5%, 10% |
| *Weber et al., 2015* | Melanoma | Nivolumab | PD-L1 | 5% |
| *Topalian et al., 2012* | Melanoma, RCC, NSCLC, CRC, PC | Nivolumab | PD-L1 | 5% |
| *Garon et al., 2015* | NSCLC | Pembrolizumab | PD-L1 | 1%, 50% |

RCC: renal cell. UCC: urothelial cell carcinoma. CRC: colorectal carcinoma. NSCLC: non-small lung carcinoma. HNSCC: head and neck squamous cell carcinoma. PC: prostate carcinoma. TIL: tumor-infiltrating lymphocytes

