## [Editor Report]

A mathematical model was established for predicting immunotherapy efficacy in this work. With three convenient available clinical parameters, the model has exhibited considerable predictive capacity with stable performance across several tumor types. It may show great promise in selecting participants for prospective trials and guiding targeted application of immunotherapy in cancer patients.

---

## [Decision Letter]

**Decision letter after peer review:**

Thank you for submitting your article "Early prediction of clinical response to checkpoint inhibitor therapy in human solid tumors through mathematical modeling" for consideration by *eLife*. Your article has been reviewed by 3 peer reviewers, and the evaluation has been overseen by a Reviewing Editor and Mone Zaidi as the Senior Editor. The reviewers have opted to remain anonymous.

Essential revisions:

1) Multiple parameters could reflect treatment efficacy of immunotherapy to cancers while three specific parameters were included in this model, the selection process and reasons for choosing these three parameters need to be described in detail, the definition of the three parameters seems to in discordant with clinical setting which requires further explanation.

2) As a heterogenous entity, different types of cancer possess distinct features, rationality to employ this model as one fit-in-all with a relatively small training cohort requires additional discussion after validated in a large-scale validation cohort.

3) As a novel model evaluation of the sensitivity and accuracy needs to be addressed, and comparison of this model with current applied parameters for predicting immunotherapy treatment efficacy requires to be supplemented.

*Reviewer #1 (Recommendations for the authors):*

The authors demonstrate a translational mathematical model dependent on three key parameters for describing efficacy of checkpoint inhibitors in human cancer. This paper describes some very interesting work. However, it remains questions that need to be addressed as follows:

1. The authors need present why they select these three parameters for describing efficacy of checkpoint inhibitors in human cancer. How about other factors?

2. How can the authors make sure that this model is suitable for all human cancer? Is the clinical tumor response dataset (n = 189 patients) enough?

3. What is the accuracy of the model?

4. I think there is no scope of this work shown in this "Introduction part".

5. Please identify the innovation point in the paper.

*Reviewer #2 (Recommendations for the authors):*

1. Several claims would need to be revised or toned down throughout the manuscript that describe the mathematical model as mechanistic and translatable. For example,

a. 'senescence markers (10) such as CD27, Tim‐3, CD57, and/or T‐cell receptor (TCR) repertoires (11)'. Only CD57 is a senescence marker among the listed proteins.

b. 'Our method may be implemented directly into clinical practice, as it relies on standard‐of‐care imaging and pathology' in Line 79,

c. 'the merit of this approach will rely on its future ability to reliably predict early individual patient response with the goal of improving personalized cancer care.'

d. Line 204-205: 'the model may be adaptable to other types of checkpoint inhibitors affecting the CTLA‐4 pathway'. Studies have described the effect of aCTLA4 therapies to be mediated in peripheral lymphoid organs rather than the tumour microenvironment. Therefore, the model that considers chemotaxis and intratumoural Ab activity will likely not apply to aCTLA4 therapies.

2. In addition to the assumptions described in the text, are the following assumptions implied:

a. The parameters µ and λ are considered constant over time, which implies that the ratio of cancer cells and immune infiltration remains the same over the treatment course.

b. Every cancer cell is assumed to have the same proliferation rate.

c. Each tumour cell has the same likelihood to be recognized by the immune system.

3. The cutoff used to split the µ and λ parameters and compare to responders and non-responders was done 'by maximizing the Youden's J statistic revealed cutoff thresholds where sensitivity'; A correction method needs to be applied to account for multiple testing correction. How does the result compare to using the median values and quantiles? How do changes in cutoff affect the Roc curve?

4. The manuscript text refers to Table S1 and supplementary figures, but supplementary information wasn't submitted for review. Unfortunately, the work on model sensitivity was not provided.

5. The work builds on a recent publication by the authors.

a. The manuscript could benefit from more information on the model, even if described previously elsewhere. For example, parameters in the methods and in Figure 1 are introduced but not described.

b. Importantly, the novelty of this study compared to the previous publication should be highlighted. The comparison with clinical and histology data appears to be the main novelty, which is why demonstrating the correlation with clinical data per patient, would be crucial. It is not clear why the calibration and validation dataset are different from the previous study. How do the estimated parameters differ?

6. The sensitivity and specificity of the λ parameter are low. Does it have an additive predictive value to the µ parameter?

7. How does the predictive value of the parameters compare to previously reported biomarkers of response such as TMB, % PDL1+ cells, T cell count? A predictive model controlling for age, stage, etc. should be implemented to account for these confounders.

8. How does the predictive value of the parameters change when the validation cohort is swapped with one of the calibration cohorts? How robust are the parameters to cross-validation? How does the predictive classification change with a different cancer type?

9. There is a lack of robust predictors of immunotherapy response due to a large number of confounder, such as heterogeneity at molecular and cellular level, immune escape, immunosenescence, etc. How does the model overcome this?

10. Figure 4 could be more clear and informative.

11. The published cohort in JITC has 72 patients but only 64 patients were included in the validation cohort, why is that so?

12. Which other pathological parameters correlate with λ and µwhen measured per patient, not per response group?

*Reviewer #3 (Recommendations for the authors):*

1) In the Figure 2, it would be helpful to show different cancer types as different panel.

2) An additional figure is favorable to show whether those key parameters were cancer type-specific or the antibody drugs.

---

## [Author Response]

Essential revisions:1) Multiple parameters could reflect treatment efficacy of immunotherapy to cancers while three specific parameters were included in this model, the selection process and reasons for choosing these three parameters need to be described in detail, the definition of the three parameters seems to in discordant with clinical setting which requires further explanation.

This point is well-taken. Mathematically, these are model-derived terms (i.e., after simplifying the initial model, composed of a system of differential equations) that allow for a closed-form solution, reducing the equation in a way that makes it possible to obtain unique solutions when only a few data points are available (a relatively common problem when using clinically-derived data), while also providing easy to interpret results and direct comparison to measurable parameters; we have now clarified this point on lines 222-225 of the revised manuscript. In fact, in the often chaotic, real world of day-to-day clinical practice, such simple indicators (or measures) are more likely to see adaptation, and it might likely (at least in part) explain the widespread success of the RECIST criteria. We also believe such simple measures (the values of which can be readily determined from stand-of-care measurements in current clinical practice) presents a major potential advantage for “translating” mathematical modeling-based predictions into clinical applications.

In order to help clarify the rationale behind this design process, we have now added the following text, lines 188-191:

“By reducing the model to this closed-form solution, we present the model in a form that combines related biological processes (see Figure 1) into only a few easy-to-interpret values, while also ensuring the ability to obtain unique solutions when only minimal data are available.”

2) As a heterogenous entity, different types of cancer possess distinct features, rationality to employ this model as one fit-in-all with a relatively small training cohort requires additional discussion after validated in a large-scale validation cohort.

Agreed: We have now added more discussions on this point, lines 474-476, 548-550, and 568-574. We have also softened the tone in line 470, by changing “demonstrates” to “suggests” that the overlaying master equation may be universally applied.

Lastly, because our calibration cohort consisted of n=189 patients and our validation cohort contained only n=64 patients, we believe that there may have been some inadvertent confusion regarding the number of patients in each cohort. This was likely due to undisciplined verbiage in the original text. We have now reworked this sentence (lines 55-58) to now read as follows:

“The model was calibrated by fitting it to a compiled clinical tumor response dataset (*n* = 189 patients) obtained from published anti-PD-1 and anti-PD-L1 clinical trials, and then validated on an additional validation cohort (*n* = 64 patients) obtained from our in-house clinical trials.”

3) As a novel model evaluation of the sensitivity and accuracy needs to be addressed, and comparison of this model with current applied parameters for predicting immunotherapy treatment efficacy requires to be supplemented.

Thank you for bringing out this essential suggestion for discussion.

First, to assess the accuracy of the model in the revision work, we have calculated sensitivity, specificity, positive predictive value (PPV), and negative predictive value (NPV); these are now provided in lines 390-400.

Moreover, we recognize that the novelty of the work presented in this work, which uses immunohistochemical (IHC) measurements to inform the model, might apparently have lost some of its central focus as originally presented. In fact, the model has already been demonstrated to perform reliably to predict patient response and even survival (e.g., PMID: 32426472), but this published work was done with a different approach (i.e., by using CT scan-based imaging data). While we have reported similar results here, the purpose of this current work is to evaluate how our new method of informing model parameters from IHC measurements compares to the previous (already established), imaging-based method. The purpose for doing so is two-fold: (i) Clinical data may often be convoluted/complex and measurements may be missing; thus, having multiple ways to inform the model (i.e., to determine model parameters) increases its ability to robustly handle a wider range of clinical scenarios; and (ii) IHC measurements may be collected at far earlier time points (relative to start of treatment) than imaging-based follow-up assessment of tumor responses; thus, this strategy potentially enables prediction tumor responses at an earlier time point than it was previously possible. We have revised the manuscript in several places to help bring this focus into greater clarity, as well as added discussion, lines 157-160, and 439-442.

Finally, we are actually unaware of any currently applied, clinically used parameters for predicting immunotherapy response, although several possible methods have been published in the last few years including transcriptomic rubrics (PMID 30127394), machine learning algorithms (PMID 33208341), genomic approaches (PMID 33307238) such as tumor mutational burden (PMID 31315901), among others (PMID 34112949**).** Instead, most “currently applied parameters” merely document tumor responses that have already occurred as opposed to predicting it a priori; these include the standard RECIST v1.1 and even the newer proposed response assessment rubrics specific for immunotherapy (iRECIST). As such, our model parameters, which can be determined in multiple ways (e.g., from early time point imaging and/or histopathology data), presents a clear advantage over the standard measures currently used in the clinic for predicting treatment outcome and patient survival. This key advantage of the methodology introduced here is now mentioned in the lines 126-127, 129-131, and 160-163 of the revised manuscript.

Reviewer #1 (Recommendations for the authors):The authors demonstrate a translational mathematical model dependent on three key parameters for describing efficacy of checkpoint inhibitors in human cancer. This paper describes some very interesting work. However, it remains questions that need to be addressed as follows:1. The authors need present why they select these three parameters for describing efficacy of checkpoint inhibitors in human cancer. How about other factors?

This point is again well-taken. The details of the full model derivation have recently been reported elsewhere (e.g., PMIDs 32426472, 33398132); however, we have now added it to the SI of this manuscript as well (Appendix 1, lines 14-103); and we have now added a reference to the derivation in the revised main text (line 192). We have also added a clarification as to why this approach was implemented in the revised main text (lines 188-191).

Additional details relative to this appropriate concern are also provided in this letter (please see Essential Revisions, point #1). We further note that, beyond those points, we have endeavored to carefully reduce the full model (wherein all relevant, key biological parameters must be included at mathematical derivation for scientific integrity) such that the model may be informed through clinically-measurable quantities. For example, how can we measure cytokine diffusion in vivo or in patients during the course of actual treatment? We cannot, at least by using the currently available technologies in a standard outpatient clinic or inpatient hospital setting. We have now found a way to solve this dilemma by further simplifying the model to reduce the parameter space of the model through biologically sound, reasonable assumptions (please see the newly added section on mathematical model derivation; Appendix 1, lines 14-103 of the revised work). That is, by focusing on only parameters that can actually be measured in current clinical practice, it allows us to focus this work on demonstrating additional ways that the model may be used for clinical predictions under current clinical practice, with a focus here on IHC measurement. We take our successful results by using only these limited data (% change in tumor volumes vs. time or IHC staining) as evidence of the robustness of the model performance.

2. How can the authors make sure that this model is suitable for all human cancer? Is the clinical tumor response dataset (n = 189 patients) enough?

The Referee has appropriately expressed some caution over the sample size of the patient cohorts. While future studies will obviously be required in order to fully address “all human cancer” as he/she inquired, we have now softened the tone of model results, clarifying that the model has only been tested in the cancer types described in the revised manuscript (lines 474-476, 548-550, and 568-574). Specifically, there were n=55 responders and n=134 non-responders. Assuming type-1 error α=0.05 and type=2 error β=0.1, the measured difference between means for μ responders vs non-responders (0.054 vs. 0.013, respectively) with population variance S^2^_μ_ = 0.00394 (data not shown in the manuscript) could be detected by a minimum necessary sample size of n=52 patients (comparison of 2 independent means). Equivalent analysis for *Λ* revealed a minimum sample size of n=51, so the calibration dataset is sufficiently large enough for the cancer types that have been reported here.

3. What is the accuracy of the model?

To assess the accuracy of the model, we have now calculated sensitivity, specificity, positive predictive value (PPV), and negative predictive value (NPV); these are now provided in lines 390-400.

4. I think there is no scope of this work shown in this "Introduction part".

We have clarified this in the revised manuscript, lines 157-160.

5. Please identify the innovation point in the paper.

We have clarified this in the revised manuscript, lines 157-160.

Reviewer #2 (Recommendations for the authors):1. Several claims would need to be revised or toned down throughout the manuscript that describe the mathematical model as mechanistic and translatable. For example,a. 'senescence markers (10) such as CD27, Tim‐3, CD57, and/or T‐cell receptor (TCR) repertoires (11)'. Only CD57 is a senescence marker among the listed proteins.

Agreed. Tim-3 is actually a marker of T-cell exhaustion, and loss of CD27 is associated with immune incompetence, exhaustion, or premature senescence (e.g., PMID: 15882352). We have now corrected this sentence to read (lines 121-124 of the revised manuscript):

“An accumulating body of evidence has established that certain immunological features, including T-cell exhaustion (e.g., Tim-3) and exclusion (9), senescence markers (10) such as CD57, or immune incompetence, exhaustion, or premature senescence (e.g., loss of CD27) (11, 12) could perhaps reflect or even predict sensitivity and resistance to checkpoint inhibitor-based cancer immunotherapy.”

b. 'Our method may be implemented directly into clinical practice, as it relies on standard‐of‐care imaging and pathology' in Line 79,

We take this to mean that, although our method could be translated to the clinic by using only already-established techniques, it must be rigorously validated in future studies, just as with any predictive clinical parameter or tool. We have added this caveat to the discussion of the revised manuscript, lines 474-476, 548-550, and 568-574.

c. 'the merit of this approach will rely on its future ability to reliably predict early individual patient response with the goal of improving personalized cancer care.'

This goes along with the previous point (b); please see our response above.

d. Line 204-205: 'the model may be adaptable to other types of checkpoint inhibitors affecting the CTLA‐4 pathway'. Studies have described the effect of aCTLA4 therapies to be mediated in peripheral lymphoid organs rather than the tumour microenvironment. Therefore, the model that considers chemotaxis and intratumoural Ab activity will likely not apply to aCTLA4 therapies.

We have recently demonstrated this to be the case (PMID 32426472). That being said, to avoid any further confusion, we have removed this statement (as well as mention of cell-based immunotherapies at this location) from the main text.

We however emphasize that this is not the intended implication of the model assumption as described in Appendix 1, lines 25-26 (that is, in our model we assume that the therapeutic cite of action occurs within the tumor microenvironment). The PD-1 and CTLA-4 receptors that are targeted in many types of immunotherapy are expressed on T cells; which may not be located in the tumor at time of systemic drug administration (while the PD-L1 receptor, located on the cancer cell plasma membrane, will always be in the tumor). This implies that, in the case of anti-CTLA-4 and anti-PD-1 therapy, a portion of the drug can potentially bind to T-cells outside the tumor (e.g., in the thymus, peripheral blood, etc.). However, our model assumes that (i) drug diffuses through the tumor to its binding site (as per Eqs. S3, S4), (ii) all drug binding occurs within the tumor, and (iii) all drug effects (e.g., tumor kill) occurs within the tumor microenvironment. That is, all “sites of therapeutic action” are located within the tumor.

We recognize that this statement may indeed be inadvertently confusing, so we have now clarified this in the model description of the revised Appendix 1 as follows (Appendix 1, lines 26-28):

“that is, because our model only describes the tumor region and the factors and processes contained therein, we have made the assumption that all key mechanisms such as drug binding occur only within the tumor.”

2. In addition to the assumptions described in the text, are the following assumptions implied:a. The parameters µ and λ are considered constant over time, which implies that the ratio of cancer cells and immune infiltration remains the same over the treatment course.

It is established that, for example, T-cells do experience “burnout” and loss of cytotoxic efficacy over their lifetimes (e.g., our reply to point 1a, above), and will influx/efflux the tumor over time, and as such it should not be expected that T-cell density remains constant in the tumor at all times. Thus, these parameters represent the average value of these fluctuating parameters. We take this to be a reasonable assumption because (i) these quantities are not directly measurable, and (ii) even in our in-house data, times between patient reassessment ranged ~4-6 weeks, so even if you could measure these, you would be forced to average between these times either way, and (iii) these processes take place at far shorter times than patient reassessments. By presenting the model in this reduced-form, we enable simplified interpretation of the results while also providing a single, easy to understand scalar that contains significant information about the treatment response. We also note here that RECIST and its immune checkpoint inhibitor counterparts provide readout in a similar manner. These points are now emphasized in the revised manuscript (lines 535-542).

b. Every cancer cell is assumed to have the same proliferation rate.

This is indeed a correct observation. Because we do not explicitly model individual cancer cells, we must use the average proliferation rate across the entire tumor in order to capture the total proliferation and proliferation rate of the cancer cell population. Also, understandably, this is not something that would even be measurable on a per-cell level in the clinic or in vivo. This approach is commonly used in mathematical modeling studies based on continuum representations. We have now pointed out this limitation in the revised manuscript (lines 537-538, with accompanying discussion in lines 535-542).

c. Each tumour cell has the same likelihood to be recognized by the immune system.

Again, this is correct, for the same reasons provided in the previous reply. Future single-cell imaging may shed light in these limitations and refine the methodology introduced here.

3. The cutoff used to split the µ and λ parameters and compare to responders and non-responders was done 'by maximizing the Youden's J statistic revealed cutoff thresholds where sensitivity'; A correction method needs to be applied to account for multiple testing correction. How does the result compare to using the median values and quantiles? How do changes in cutoff affect the Roc curve?

We appreciate the attention to detail. We believe the suggestion for multiple testing correction here is likely based on a potential misunderstanding. We did not identify an “ideal” threshold for each cohort by Youden’s J statistic, we only identified one single threshold for *μ* and *Λ* (that is, one for each parameter) in the calibration cohort, and then assessed the performance of the same threshold values in the validation cohorts. The results show that the thresholds were able to satisfactorily separate responders from non-responders, which in our opinion has proven the validity of *μ* and *Λ* for this purpose already, i.e., to test how these carefully designed thresholds would perform across independent datasets. Therefore, it is a little unclear as to what value testing other cutoff thresholds would add to the manuscript. However, in order to avoid any confusion, we have clarified this in the text, lines 392-393:

“Testing these same response threshold values (identified in the calibration cohort) in the validation cohort”.

4. The manuscript text refers to Table S1 and supplementary figures, but supplementary information wasn't submitted for review. Unfortunately, the work on model sensitivity was not provided.

We have now verified that we had originally uploaded both main text and SI at time of initial submission. We do not know why the SI (now Appendix 1) has not been properly provided to the Referees for external peer-review, but regret that this may have indeed been the case. We will be sure to submit both revised main text and SI when we resubmit the revised draft; we will also check with the editorial office to make sure that they have been fully transmitted to the Referees.

5. The work builds on a recent publication by the authors.a. The manuscript could benefit from more information on the model, even if described previously elsewhere. For example, parameters in the methods and in Figure 1 are introduced but not described.

Agreed. We have added a detailed model derivation, including discussion of individual parameters and assumptions made, in Appendix 1 (Appendix 1, lines 14-103); the reader is also now referred to this on line 192.

b. Importantly, the novelty of this study compared to the previous publication should be highlighted. The comparison with clinical and histology data appears to be the main novelty, which is why demonstrating the correlation with clinical data per patient, would be crucial. It is not clear why the calibration and validation dataset are different from the previous study. How do the estimated parameters differ?

Thank you for this suggestion. Unfortunately, we have been unable to obtain any per-patient data that contains both patient response to ICI therapy and also IHC measures of PD-L1 expression or CD8^+^ tumor-infiltrating lymphocyte counts from the same patients that would enable this direct comparison. While we are working towards in-house collection of this data, we believe that this study represents an important step towards this goal. The calibration cohort in this work was expanded from the literature cohort used in previous work (PMID: 32426472), because we found more published data since that time, and wanted to use all data available. Model parameters are the same (per-patient) between studies when estimated by tumor volume measurements (although means have changed due to addition of new data); these serve as the established baseline for comparison with the IHC-based estimation of model parameters presented in this work.

We however fully agree that further emphasis should be placed on the novelty of informing model parameters by IHC; therefore, we have further emphasized this point in the manuscript, lines 157-160 and 439-442.

6. The sensitivity and specificity of the λ parameter are low. Does it have an additive predictive value to the µ parameter?

It is correct that µ was found to have higher sensitivity and specificity in the results shown. We have now included a full description of the results for *Λ* as well, (i) for sake of scientific completeness, because it is an integral part of the model reduced-form and we intended to show the complete set of our results, and (ii) because we have shown previously (PMID: 33398132) that *Λ*×*μ* may also be used as a dug-disease specific measure of treatment response.

7. How does the predictive value of the parameters compare to previously reported biomarkers of response such as TMB, % PDL1+ cells, T cell count? A predictive model controlling for age, stage, etc. should be implemented to account for these confounders.

We were unable to obtain TBM, % PDL1+ cells, or intratumoral T cell count for the patients studied. This was not in the literature-derived cohort, and needle biopsies were not required in the in-house studies, so we have been as yet unable to do these analyses at this time. Age, stage, etc. are well suited for certain kinds of statistical models (e.g., Cox proportional hazards), and we agree that this could be something interesting to investigate, e.g., maybe we could find out whether the model could serve better for age groups of patients. That said, since this is not the focus of this work, we have decided to put it as part of our future work, lines 568-574:

" Towards this end, we are currently pursuing in-house collection of per-patient, paired IHC measures of PDL1 or T-cell counts with tumor response, which will enable us to directly correlate IHC measures of interest with tumor response, while also allowing for examination of additional patient parameters, such as tumor stage or patient age.”

8. How does the predictive value of the parameters change when the validation cohort is swapped with one of the calibration cohorts? How robust are the parameters to cross-validation? How does the predictive classification change with a different cancer type?

This point is well-taken. In fact, we have shown similar results to the analysis requested here previously, in PMID: 32426472, Figure 5. In order to shed further light on parameter robustness across cancer types, as suggested, we have added a new analysis wherein we performed a ‘leave-one-cancer-type-out’ validation analysis with the calibration cohort (made up of 6 different cancer types). That is, the model parameter rages for each response category from the remaining 5 cancer types are compared to the range from the left-out type; this was repeated for all 6 cancer types, and the results are shown in the new figure, Appendix 1-figure 3. We have also added discussion on this new analysis in the main text, lines 356-361. We have also now referred the reader to this publication (PMID: 32426472) in case they are interested, line 360. We would like to thank the Referee for his/her helpful suggestion.

9. There is a lack of robust predictors of immunotherapy response due to a large number of confounder, such as heterogeneity at molecular and cellular level, immune escape, immunosenescence, etc. How does the model overcome this?

This is a fair point, and the Referee is correct that these cofounding factors are known to be relevant in ICI treatment and outcome. In the model as presented, we approximate heterogeneous parameters (such as immune cell distribution or infiltration rates, immune checkpoint ligand densities, ICI drug perfusion, immune cell activation and killing capacity, etc.) based on their spatiotemporal averages. As such, we overcome these by indirectly capturing their contributions to the overall system, allowing us to characterize the system in order to make predictions without explicit measures of these factors.

Indeed, complete characterization of these factors remains clinically prohibitive, if even possible, and we believe that the high cost (time, money, and inconvenience of clinical adoption) of such models that must capture them to function will not progress beyond the research or hypothesis-generating stage. We also believe that these low-level factors contribute to overarching mechanisms that are more readily measurable, allowing their effects (including those not as yet discovered) to be captured indirectly. By developing methods to inform more parameters in our model by using clinically-available measures, we are moving forward to a place where one day we may be able to mathematically back-calculate these unmeasurable factors, which we hope will be of benefit to both researchers and clinicians.

10. Figure 4 could be more clear and informative.

If the Referee could perhaps help us by further clarifying his/her comment, that would be greatly appreciated; we are open for constructive suggestions.

11. The published cohort in JITC has 72 patients but only 64 patients were included in the validation cohort, why is that so?

Some patients were excluded due to inclusion criteria specific to this study, which were different than those in JITC. Details are now provided on lines 263-266 of the revised manuscript:

“a total of 95 patients were obtained; however, 18 left the study after admission, 11 were removed due to lack of pre-treatment measurements, one was excluded because all indexed lesions were treated with XRT, and one discontinued treatment but continued follow-up”.

12. Which other pathological parameters correlate with λ and µwhen measured per patient, not per response group?

Thank you for the comment. Unfortunately, only % change in tumor burden over time was available for our calibration cohort; thus, we were unable to investigate pathological parameters in this study. Please also see our reply to Referee #2 Public Review comment 2, above.

Reviewer #3 (Recommendations for the authors):1) In the Figure 2, it would be helpful to show different cancer types as different panel.

We appreciate this comment; however, we elected this format to simply highlight the differences in parameter values while keeping our focus on the main innovation of this revised manuscript: i.e., by using IHC to inform model parameters. We have shown results similar to those requested here in a previous study however; we have now added text to point the interested reader to these results, line 771.

2) An additional figure is favorable to show whether those key parameters were cancer type-specific or the antibody drugs.

Please see the previous comment. We have shown cancer-drug specific results in our previous work; we have added a reference to point the interested reader to this analysis, line 771.